# A neural circuit model for human sensorimotor timing

Seth W. Egger 🔘 [1,2,4,5✉], Nhat M. Le 🔘 [2,3,5] & Mehrdad Jazayeri 🔘 [1,2✉]

Humans and animals can effortlessly coordinate their movements with external stimuli. This capacity indicates that sensory inputs can rapidly and flexibly reconfigure the ongoing dynamics in the neural circuits that control movements. Here, we develop a circuit-level model that coordinates movement times with expected and unexpected temporal events. The model consists of two interacting modules, a motor planning module that controls movement times and a sensory anticipation module that anticipates external events. Both modules harbor a reservoir of latent dynamics, and their interaction forms a control system whose output is adjusted adaptively to minimize timing errors. We show that the model's output matches human behavior in a range of tasks including time interval production, periodic production, synchronization/continuation, and Bayesian time interval reproduction. These results demonstrate how recurrent interactions in a simple and modular neural circuit could create the dynamics needed to control timing behavior.

[1] McGovern Institute for Brain Research, Cambridge, MA, USA. [2] Department of Brain and Cognitive Sciences, Cambridge, MA, USA. [3] Picower Institute for Learning and Memory, Massachusetts Institute of Technology, Cambridge, MA 02139, USA. [4] Present address: Department of Neurobiology, Duke University School of Medicine, Durham, NC 27710, USA. [5] These authors contributed equally: Seth W. Egger, Nhat M. Le. ✉email: swegger@mit.edu; mjaz@mit.edu

Sensorimotor coordination in humans is remarkably flexible. We can anticipate events based on few observations and use that information to adjust our movements. For example, musicians can use a metronome to adjust the tempo of their movements, and children can rapidly coordinate their movements during a clapping game. However, we still lack an understanding of how networks of neurons generate such coordinated movements.

Recent studies have proposed that the neural basis of sensorimotor coordination may be understood using the language of dynamical systems[1–3]. The key intuition is that recurrent neural networks in the motor cortex form dynamical systems[4,5] whose output can be controlled by sensory inputs[2,6–9]. This idea has been explored in large-scale distributed recurrent neural network models[10]. However, the complexity of these network models often makes it difficult to understand their behavior from an algorithmic perspective.

Timing provides a prime example of sensorimotor coordination and is crucial in behaviors demanding the generation of delayed motor responses, generation of rhythmic movements with a desired tempo, and synchronization of movements to anticipated events. An early model proposed that the brain controls action timing by adjusting the speed of an internal clock whose ticks are integrated toward a fixed threshold[11]. Consistent with this proposal, experiments in animal models have found that neural activity in anticipation of a delayed response reaches a fixed threshold[12–14] at a rate that is inversely proportional to the delay period[7,15,16]. These results suggest that the brain supports flexible timing by controlling the speed at which neural activity approaches a movement initiation threshold.

Recently, it was shown that flexible control of speed can be achieved through nonlinear interactions within a simple model consisting of a pair of units with reciprocal inhibitory connections[7]. In this model, the speed at which the output evolves toward a movement initiation threshold can be adjusted flexibly via a shared input (Fig. 1a). From a dynamical systems perspective, this model can be viewed as an open-loop controller that converts an instruction (i.e., shared input) to the desired dynamics (i.e., speed).

However, the larger utility of this model depends on whether it can be extended to accommodate additional constraints associated with temporal control of movements. In particular, temporal control of movements must accommodate noise in the nervous system[17], prior expectations[18], and sensorimotor delays imposed by internal[19] and external[20] temporal contingencies. These factors limit the capacity of open-loop systems to achieve robust temporal control. Efficient control systems often rely on sensory feedback to combat noise, and when sensory feedback is delayed, they rely on a mechanism to predict future sensory and motor states[19,21,22]. Here, we augment the original open-loop system with a sensory anticipation module (SAM) and a feedback mechanism, and show that the resulting neural circuit model can accommodates internal noise, prior expectations, and sensorimotor delays. Finally, we demonstrate that the model can capture key features of human behavior in a number of classic timing tasks (Fig. 1b–e).

## Results

We will describe the full model in four steps. We start by introducing a basic circuit module (BCM) that acts as a flexible open-loop controller for producing desired time intervals. We extend the BCM to a motor planning module (MPM) capable of producing isochronous rhythms. We then introduce a SAM that provides the means for anticipating and predicting upcoming temporal events. Finally, we introduce the full model that combines the MPM and SAM to create a system that can dynamically coordinate motor plans and actions with anticipated and unanticipated stimuli in a range of behavioral tasks.

**BCM for interval production.** The BCM has been described in detail previously[7]. Briefly, the BCM includes three units, $u$, $v$, and $y$, each representing the average activity of a population of neurons (Fig. 1a, top). $u$ and $v$ inhibit one another and receive input, $I$, that is tonic, or constant over time. $y$ receives excitatory input from $u$ and inhibitory input from $v$, and leverages the nonlinear dynamics of mutual inhibition between $u$ and $v$ to generate ramp-like activity. Finally, the model initiates a "movement" when $y$ crosses a fixed threshold, $y_0$. The rate dynamics of $u$, $v$, and $y$ are as follows:

$$\tau \frac{du}{dt} = -u + \theta(W_{uI}I - W_{uv}v + \eta_u), \quad (1)$$

$$\tau \frac{dv}{dt} = -v + \theta(W_{vI}I - W_{vu}u + \eta_v), \quad (2)$$

$$\tau \frac{dy}{dt} = -y + W_{yu}u - W_{yv}v + \eta_y. \quad (3)$$

$W_{uI}$ and $W_{vI}$ denote the strength with which $I$ drives $u$ and $v$, respectively. $W_{uv}$ and $W_{vu}$ denote the strength of inhibitory coupling from $v$ to $u$, and from $u$ to $v$, respectively. $\tau$ is the time constant of each unit. $\theta(x)$ is a sigmoidal function that maps the input to an output between 0 and 1 (see "Methods"). Finally, $\eta_u$, $\eta_v$, and $\eta_y$ are stochastic synaptic inputs to each unit and are modeled as independent white noise with standard deviation $\sigma_n$.

We assume $W_{uI} = W_{vI} = 6$ (identical shared excitatory input), $W_{uv} = W_{vu} = 6$ (symmetric mutual inhibition), $W_{yu} = W_{yv} = 1$, and $\tau = 100$ ms for all units, a value consistent with previous models. With these parameters, the model functions in a dynamical regime with three fixed points: an unstable point at $u = v$, and two stable fixed points with either $u$ dominating or $v$ dominating (Fig. 1a, middle). In this regime, $u$ and $v$ evolve, or change over time, toward one of the stable fixed points depending on the initial conditions. These dynamics lead to a ramp-like activity in $y$ whose rate is inversely related to $I$ (Fig. 1a, bottom). In other words, the BCM acts as an open-loop controller that converts an instruction conveyed by $I$ to a dynamic output, $y$, that evolves toward the threshold, $y_0$, at an appropriate speed (Fig. 1a). Therefore, by adjusting $I$, the BCM can flexibly adjust the movement initiation time.

**MPM for periodic production.** We next considered the production of multiple movements with a fixed inter-production-interval (IPI). This is challenging with the BCM which can only produce a single action. One solution is to use multiple concatenated BCMs, but that seems unrealistic as the number of modules would grow with the number of movements. We tackled this problem using a modification of the BCM, which we call the MPM. The MPM is identical to BCM except for an additional reset mechanism after each movement that allows the model to generate an arbitrary number of timed outputs (Fig. 2a). From a neurobiological perspective, the reset signal can be generated by the corollary discharge associated with the movement command[23,24], or by sensory feedback triggered by the movement. We implemented the reset as a transient 10 ms pulse, $I_p$, activated right after the threshold crossing (Fig. 2a, dashed line). Mathematically, this can be formalized as follows:

$$\tau \frac{du_p}{dt} = -u_p + \theta\left(W_{u_pI}I - W_{u_pv_p}v_p + \eta_{u_p} - I_p\right), \quad (4)$$

$$\tau \frac{dv_p}{dt} = -v_p + \theta\left(W_{v_pI}I - W_{v_pu_p}u_p + \eta_{v_p} + I_p\right). \quad (5)$$

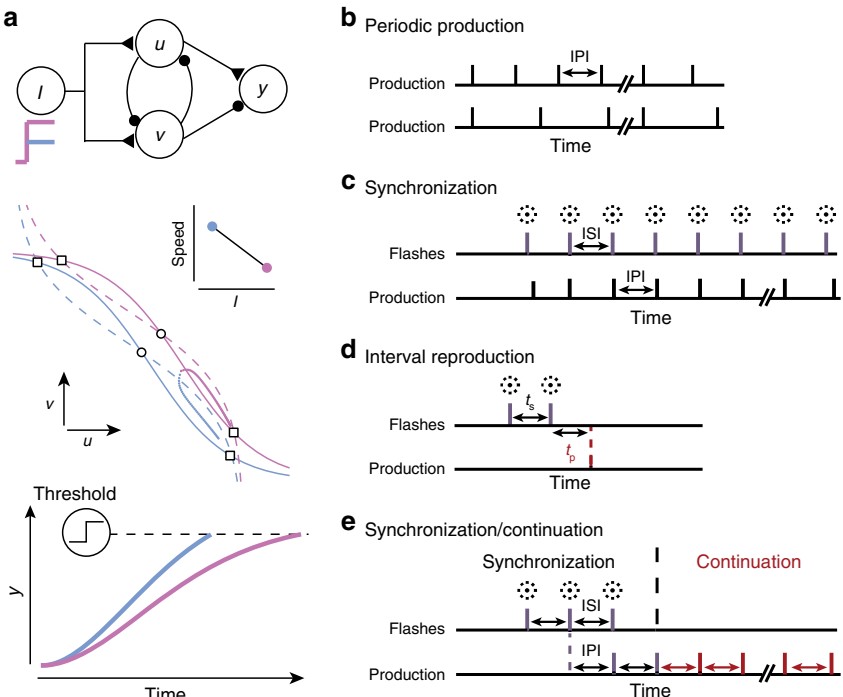

**Fig. 1 Basic circuit module (BCM) architecture and timing behavior. a** BCM architecture (top) consists of two units, $u$ and $v$, that inhibit each other. $u$ and $v$ receive common input, $I$ (colored step functions), and drive an output unit, $y$, with excitatory and inhibitory connections, respectively. Excitatory and inhibitory connections are shown by triangles and circles, respectively. The speed control mechanism can be understood by analyzing system dynamics in the phase plane of $u$ and $v$ (middle; after Wang et al.[7]). Dots represent $u$ and $v$ over time for a large (pink) or small (blue) input for no noise and with initial conditions set so that that $u$ eventually dominates $v$. The system's saddle and stable fixed points are indicated by larger circles and squares, respectively. Dashed and solid lines indicate the nullclines of $u$ and $v$, respectively. Inputs configure the positions of the nullclines for $u$ and $v$, and therefore, control the speed. Inset shows the relationship between speed and input. Because $u$ excites and $v$ inhibits $y$, operation of the system in this regime increases $y$ (bottom) until it reaches a threshold for action initiation (dashed line). **b**–**e** Classic timing tasks used to study human timing behavior. **b** Periodic production requires the subject to produce a series of actions over time (vertical lines), with a constant inter-production-interval (IPI). Top and bottom are examples of two different IPIs. **c** Synchronization requires the subject to time a series of actions (vertical black lines) such that they are simultaneous with a series of sensory inputs (e.g. flashes; vertical magenta lines) with a set inter-stimulus-interval (ISI). **d** Interval reproduction requires the subject to measure an interval, $t_s$, demarcated by two stimuli (flashes; vertical magenta lines) and to produce an interval, $t_p$, by initiating an action (dashed red line). $t_p$ has to match $t_s$ as accurately as possible. **e** Synchronization/ continuation requires the subject to synchronize actions to a series of inputs with an ISI selected at random from a prior distribution and then, after the stimulus is extinguished, continue to produce actions with an IPI matching the ISI selected on a given trial.

In the MPM, $u_p$, $v_p$, and $y_p$ evolve identically to the BCM between consecutive threshold crossings (Fig. 2b). In addition to producing an output at each threshold crossing, however, the model activates $I_p$, which resets $u_p$ and $v_p$, and causes a rapid drop in $y_p$. This allows the circuit to restart the dynamics and generate another output. As expected, IPIs increase monotonically with $I$ (Fig. 2c, d; $r^2 = 0.84$; $F(1, 160) = 828.6$; $p \ll 0.01$) within a suitable range of inputs (Supplementary Fig. 1).

In humans, IPIs are variable with a standard deviation that increases linearly with the mean[25]. To evaluate IPI variability in the model, we performed simulations in the presence of Gaussian noise (see "Methods"). When noise levels were not too high (Supplementary Fig. 2), the model exhibited a qualitatively similar behavior to that of humans (Fig. 2e): IPI variability increased monotonically with IPI (one-tailed F test; $I = 0.75$ to $I = 0.76$: $p = 0.014$, $F(80, 70) = 0.60$; $I = 0.76$ to $I = 0.77$: $p < 0.01$, $F(57, 70) = 2.95$; $I = 0.77$ to $I = 0.78$: $p < 0.01$, $F(40, 57) = 10.93$). However, this relationship was nonlinear in the model, which is unsurprising given the simplicity of our assumptions regarding the nature of noise in the brain and the underlying dynamics.

**SAM for predicting future events.** In many circumstances, the IPI is not known in advance and has to be adjusted based on

external timing cues. Here, we considered a case in which the IPI has to be adjusted by the interval between external temporal events. In general, measuring time between events can be achieved in two ways. One way is to implement a circuit whose output is a ramp with a fixed slope. In such a circuit, the output level provides a moment-by-moment estimate of elapsed time (Supplementary Fig. 3a). Alternatively, the circuit may function predictively, and adjust the slope of the ramp such that the output reaches a certain expected level at the anticipated time of the next event (Supplementary Fig. 3b).

A recent study in monkeys found evidence in support of the predictive mechanism[20]: neural dynamics in the frontal cortex were adjusted so that responses reached an expected state at the expected time of the stimulus. Based on this finding, we developed a SAM that measures time predictively (Fig. 3a). The SAM behaves identically to the MPM except that it does not generate an action when its output, $y_s$, reaches $y_0$. Instead, the SAM adjusts the input so that $y_s$ would reach $y_0$ exactly at the expected time of the sensory feedback. When $I$ is too high, $y_s$ would increase at a slower pace than needed, and its value at the time of the sensory feedback would be below $y_0$. Conversely, when $I$ is too low, $y_s$ would go beyond $y_0$ at the time of the feedback. Accordingly, the system must decrease $I$ when $y_s < y_0$, and increase $I$ when $y_s > y_0$. The SAM implements these

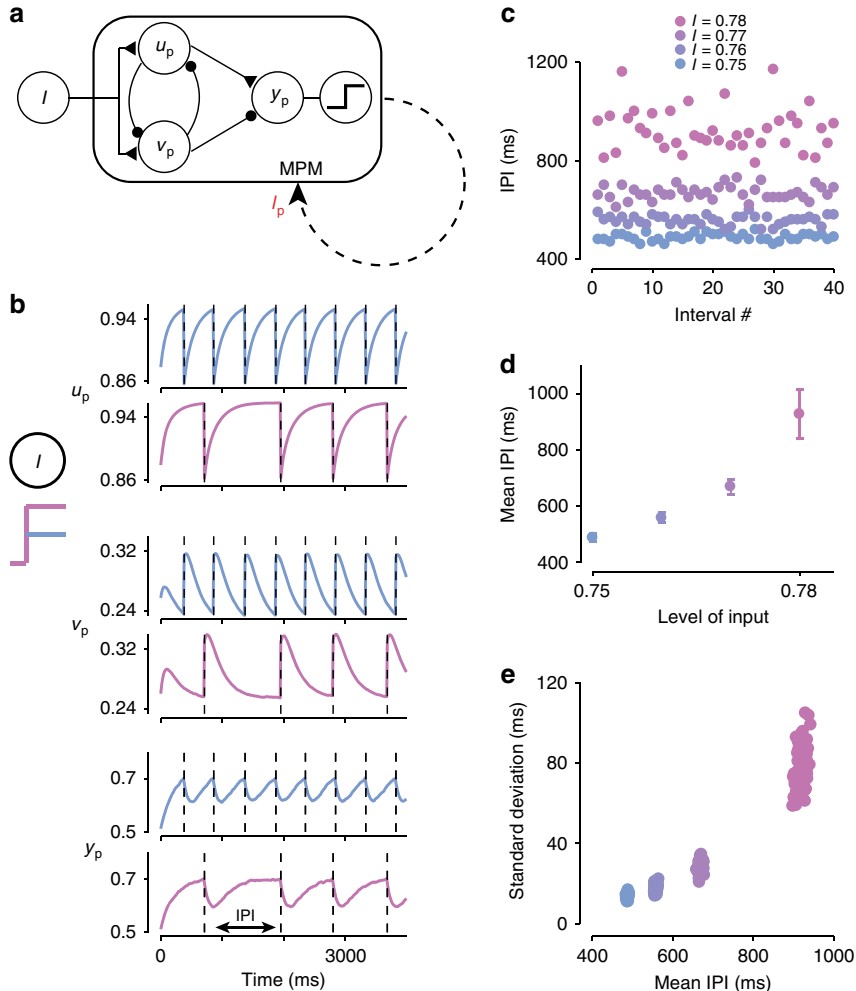

**Fig. 2 Input to the motor planning module (MPM) controls frequency of periodic production. a** The architecture of the MPM is identical to the BCM, with the addition of a reset mechanism that is activated when $y_p$ crosses the threshold (represented by the nonlinear unit at right), and causes the output to be fed back as an input, $I_p$ (dashed line). Excitatory and inhibitory connections are shown by triangles and circles, respectively. **b** Behavior of the units $u_p$, $v_p$ and $y_p$ in response to a small (blue) or large (pink) input, $I$. Vertical dashed lines indicate the timing of threshold crossing and motor output. The time between successive motor outputs is defined as the inter-production-interval (IPI). **c** Value of the first 40 IPIs generated by the MPM at each level of input, $I$ (colors), in representative trials. **d** Mean of the IPIs (±standard deviation; $N = 40$) in panel (**c**) as a function of input, $I$. **e** Standard deviation of IPIs plotted against the mean IPI for 100 trials of each level of input in the presence of Gaussian noise ($\sigma_n = 0.01$). Each data point represents the results of a trial as in panels (**c**) and (**d**).

adjustments by updating the value of $I$ dynamically at a rate proportional to the error signal, $y_s - y_0$:

$$\tau \frac{dI}{dt} = sK(y_s - y_0). \tag{6}$$

In this update rule, $K$ scales the rate of change of $I$ with the error signal, and $s$ is a gating parameter that is set to 1 when feedback is on and 0 otherwise. This formulation allows the model to update $I$ only when the feedback is on ($s = 1$). Fig. 3b illustrates the response of the SAM to three equidistant sensory inputs with a short (400 ms) or long (1000 ms) inter-stimulus-interval (ISI). The first stimulus (S1) triggers a transient input, $I_s$. This input does not alter $I$ since the system has not yet measured the ISI, but resets $u_s$ and $v_s$ (Fig. 3a; see "Methods"). Following S1, the output, $y_s$, increases over time. For a short ISI, $y_s$ at the time of the second stimulus (S2) is lower than $y_0$ causing $I$ to decrease (Fig. 3b, left). In contrast, for a long ISI, $y_s$ is greater than $y_0$ causing $I$ to increase (Fig. 3b, right). The transient input, $I_s$, triggered by S2 also resets $u_s$ and $v_s$, allowing the dynamics to

evolve with the updated speed (Fig. 3b). By iterating this process after each sensory input, the SAM dynamically adjusts its output such that $y_s$ will eventually match $y_0$ at the precise time of each stimulus input. These results are robust with respect to when the first stimulus is presented (750 ms in Fig. 3b and subsequent simulations; see Supplementary Fig. 4e for other delays).

Using simulations of the SAM, we analyzed the effect of $K$ and $I_0$ on the root-mean-squared-error (RMSE) between the predicted time (the time at which $y_s$ crosses $y_0$) and the actual time of the stimuli (Fig. 3c; see "Methods" for simulation details). When $I_0$ is large and $K$ is small, the SAM generates large errors. Intermediate values of $I_0$ and $K$ lead to smaller errors, and a specific combination of $(K^*, I_0^*)$ minimizes the RMSE (see Fig. 3c). Exploration of the parameter space revealed three key results. First, $I_0^*$ is largely independent of noise level and the number of stimuli (Fig. 3d, left). This is consistent with $I_0$ serving as an initial estimate before any feedback is integrated. Second, $K^*$ decreases with increasing $\sigma_n$ (Fig. 3d, right). This is expected because, when internal noise increases, the error signals generated by the SAM are less reliable and should be appropriately

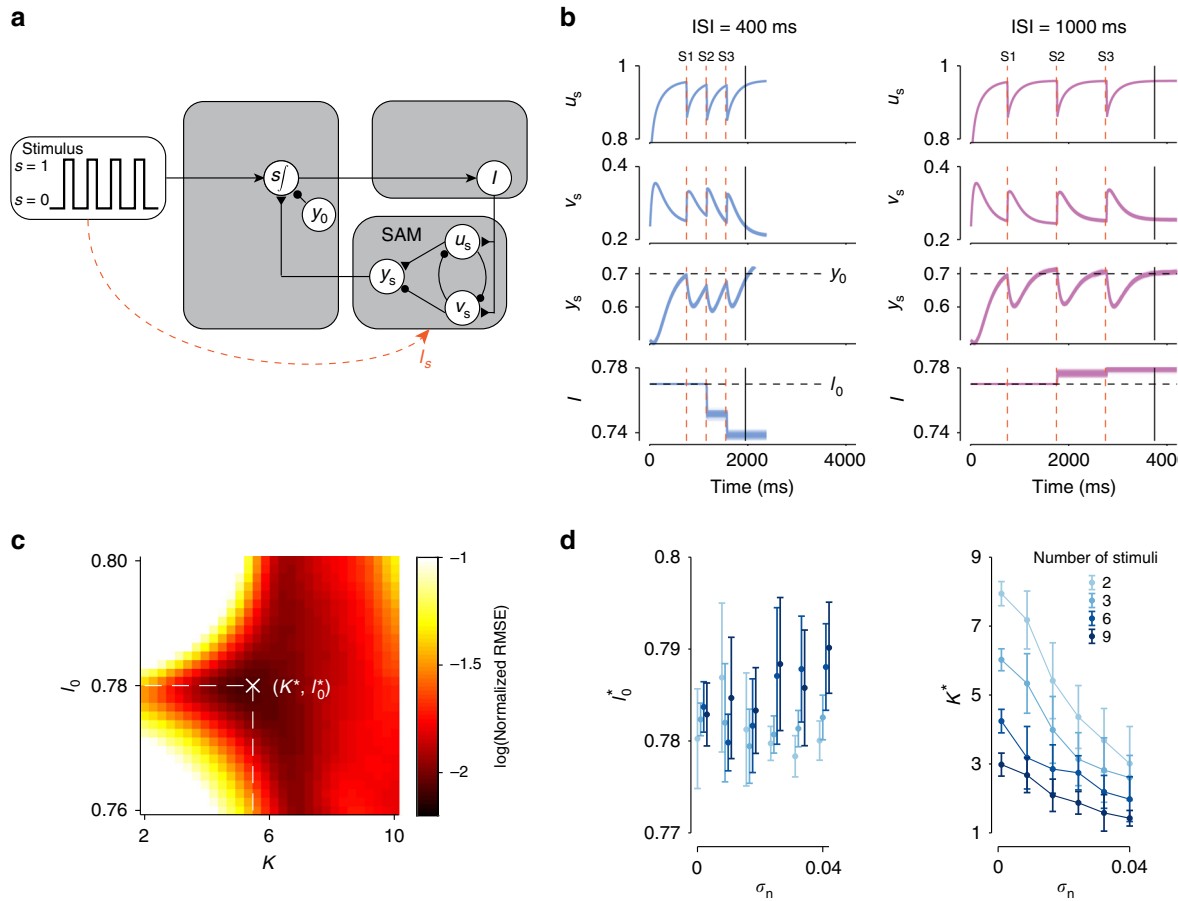

**Fig. 3 Dynamic adjustment of input with the sensory anticipation module (SAM). a** Architecture of the updating mechanism and SAM. The updating mechanism, represented by s∫, relies on the difference between the output of the SAM, $y_s$, and the desired level of activity at the time of each stimulus. This is implemented by integrating the summed activity of $y_s$ and a tonically firing inhibitory unit, $y_0$, into the input unit, $I$, when the stimulus is on ($s = 1$). When the stimulus is off ($s = 0$), integration is prevented and $I$ remains constant. Dashed line indicates a stimulus input, $I_s$, directly to $u_s$ and $v_s$ when $s = 1$, resetting the SAM after each stimulus. Excitatory connections are indicated by triangles, inhibitory connections are indicated by circles, and non-specific synapses are indicated by arrows. **b** Response of the SAM units $u_s$, $v_s$, $y_s$, and $I$ to three equidistant stimuli (S1, S2, S3, vertical dashed lines) with an ISI of 400 ms (blue) and 1000 ms (pink) with $K = 5.0$, $I_0 = 0.77$, $\sigma_n = 0.01$. The vertical black line indicates the time of the stimulus which is to be predicted. The four panels show how the activity of the four units $u_s$, $v_s$, $y_s$, and $I$, changes with time in different simulation runs. Each panel contains 100 superimposed lines, each corresponding to the activity of that unit in a different trial. The horizontal dashed lines indicate the threshold $y_0$ (bottom middle panel) and $I_0$ (bottom panel). **c** Example optimization of the value of the weight given to the update, $K$, and $I_0$, at noise level $\sigma_n = 0.005$ and $N = 3$ stimuli. Color scale represents the RMSE for each pair of $K$ and $I_0$ tested, based on 100 simulated trials for each ISI and $\sigma_n = 0.005$. **d** The optimized parameter values, $K^*$ and $I_0^*$, as a function of the level of noise, $\sigma_n$, and number of input stimuli (colors). Centers and error bars indicate the mean and standard deviation across $n = 10$ optimization runs.

discounted. Third, as $N$ increases, $K^*$ decreases, which reduces the weight given to each error and allows the model to integrate across inputs (Fig. 3d, right). Finally, we verified that the SAM updating process is robust with respect to the units' initial conditions ($u_0$ and $v_0$) (Supplementary Fig. 4a–d), and can be generalized to different dynamical regimes (Supplementary Fig. 5). Therefore, the SAM provides a plausible mechanism for measuring time intervals predictively.

**Sensorimotor updating by combining the SAM and MPM.** So far, we found that the MPM can produce different IPIs depending on the level of input, and the SAM can adjust the input level based on the anticipated time of sensory events. Accordingly, we reasoned that the SAM and MPM might together be able to generate timed outputs that are in register with incoming sensory events: the SAM would adjust the input based on sensory events and the MPM would use that input to adjust IPI. We therefore connected the SAM to the MPM by having them share the input,

$I$ (Fig. 4a), and measured IPI as we changed the ISI randomly in a blocked fashion (Fig. 4b). The circuit was able to successfully track ISI throughout the run (Fig. 4c; see "Methods"). After each block transition, the SAM detects the error between IPI and ISI and adjusts $I$ so that the MPM can gradually bring IPI closer to the new ISI (Fig. 4c). This mechanism allows IPIs to increase lawfully with the ISI ($r^2 = 0.53$; $p \ll 0.01$; Fig. 4d).

**Full circuit model for synchronization.** Coupling between the MPM and SAM allowed the circuit to adaptively match its output frequency (IPI) to the input frequency (ISI). However, it failed to match the input and output in terms of phase, as evidenced by the uniform distribution of phase differences between inputs and outputs (Fig. 4e; Rayleigh test of uniformity, $p = 0.10$). The source of this problem is that the circuit has no mechanism to determine whether the MPM output leads or lags the stimulus. Previous models have proposed highly nonlinear systems for phase adjustment[26–29]. We found that our model provides a

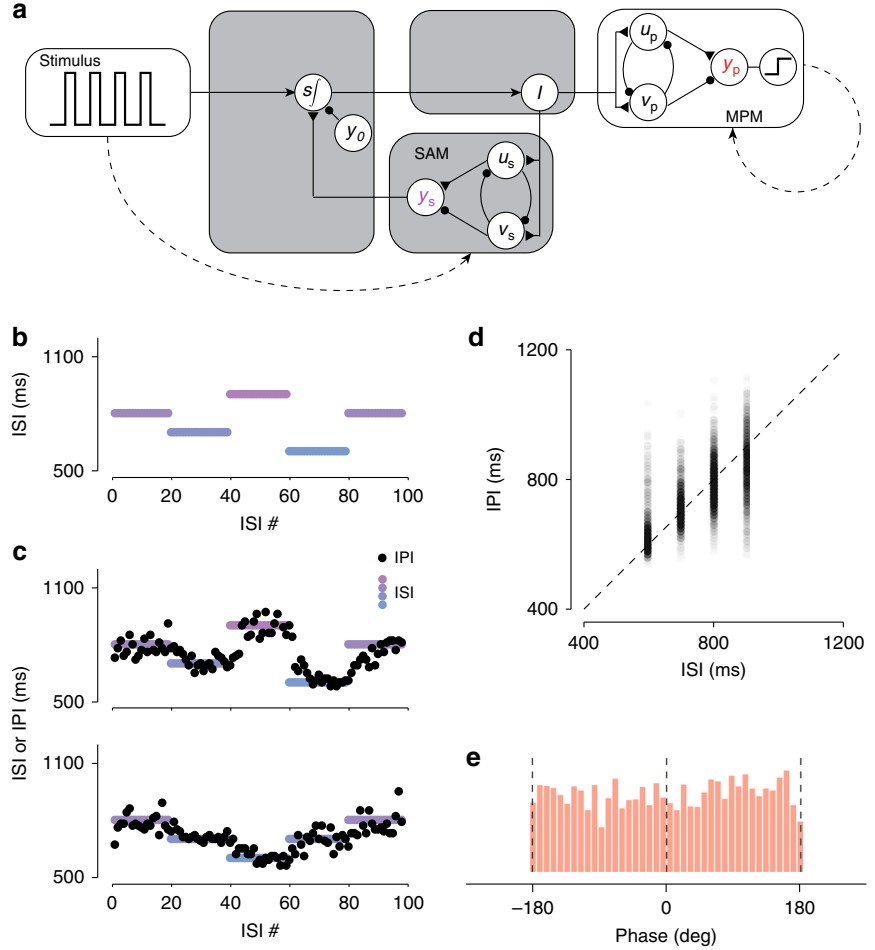

**Fig. 4 Online adjustment of inputs allows a combined circuit to match IPI to stimulus ISI. a** Wiring diagram of the circuit model that combines the sensory anticipation module (SAM) with the motor planning module (MPM). Conventions as in Fig. 2 and 3. **b** Example inter-stimulus-interval (ISI) sequence for a trial of the ISI tracking task. Colors indicate different ISIs. We initiated each trial with a block of twenty consecutive 800 ms ISIs. Each subsequent block of 20 ISIs was randomly selected from a discrete uniform distribution between 600 and 900 ms. Each trial consisted of 5 blocks and 100 total ISIs. **c** Inter-production-intervals (IPIs; black circles) associated with each ISI (colors) for two example trials of the ISI tracking task. **d** Density of IPIs (grayscale) as a function of the associated ISI. Dashed line indicates perfect tracking. **e** Distribution of the phase of motor output. A phase of 0 indicates output that is synchronous with the stimulus input.

simple solution to this problem. Since the output of the SAM ($y_s$) and MPM ($y_p$) are synchronized to sensory and motor events, respectively, the difference between them provides a graded signal reflecting their relative phase. Specifically, when $y_s > y_p$, the MPM is lagging and should be sped up, and when $y_s < y_p$, the MPM is leading and should be slowed down. To implement this solution, we augment the input $I$ with a signal $\Delta I$ of the form

$$\Delta I = \alpha(y_p - y_s), \qquad (7)$$

where $\alpha$ controls the learning rate. This adjustment can be realized by a unit which receives excitatory and inhibitory input form $y_p$ and $y_s$, respectively (Fig. 5a, cyan). Fig. 5b demonstrates this correction scheme. Initially, $y_p$ slightly lags behind $y_s$. This asynchrony generates a biphasic $\Delta I$ whose value is transiently positive (between the stimulus onset and motor output) and then negative until the next stimulus onset. The addition of this error signal reduces the total tonic drive to the MPM and increases the output speed relative to the SAM. After this adjustment is applied over several ISIs/IPIs, the MPM becomes increasingly synchronized with the stimulus. This strategy allows the circuit to gradually reduce asynchrony and narrow the distribution of phase errors (Rayleigh test of uniformity, $p \ll 0.01$; Fig. 5c).

We examined the behavior of the model for different values of $\alpha$ and $K$. Larger values of $\alpha$ make the model more sensitive to phase errors and allow the model to cancel asynchronies more effectively (Fig. 5d, left panel). However, larger values of $\alpha$ cause rapid beat-by-beat adjustments in IPI, which can compromise rhythmicity (Fig. 5d, middle panel). As such, the optimal value of $\alpha$ depends on the relative cost of minimizing phase asynchrony and IPI error. In our simulations, values between 0.1 and 0.15 allow the model to minimize the sum squared error of phase and IPI (Fig. 5d, right panel). The parameter $K$, on the other hand, has little effect on how $\alpha$ influences phase asynchrony and IPI error (Fig. 5d, different colors).

This mechanism allowed the model to capture two counter-intuitive observations in human behavior. First, as shown in Fig. 5c, the MPM had a persistent phase lead relative to the stimulus ($-27.14° \pm 71.45°$) in a manner similar to humans performing analogous tasks[30]. Second, the interaction of $\alpha$ with the level of noise caused the model to occasionally skip a beat (Supplementary Fig. 6), which occurs occasionally in humans performing similar tasks[31].

**Model responses to perturbations are similar to humans**. We compared the behavior of the model to that of humans in

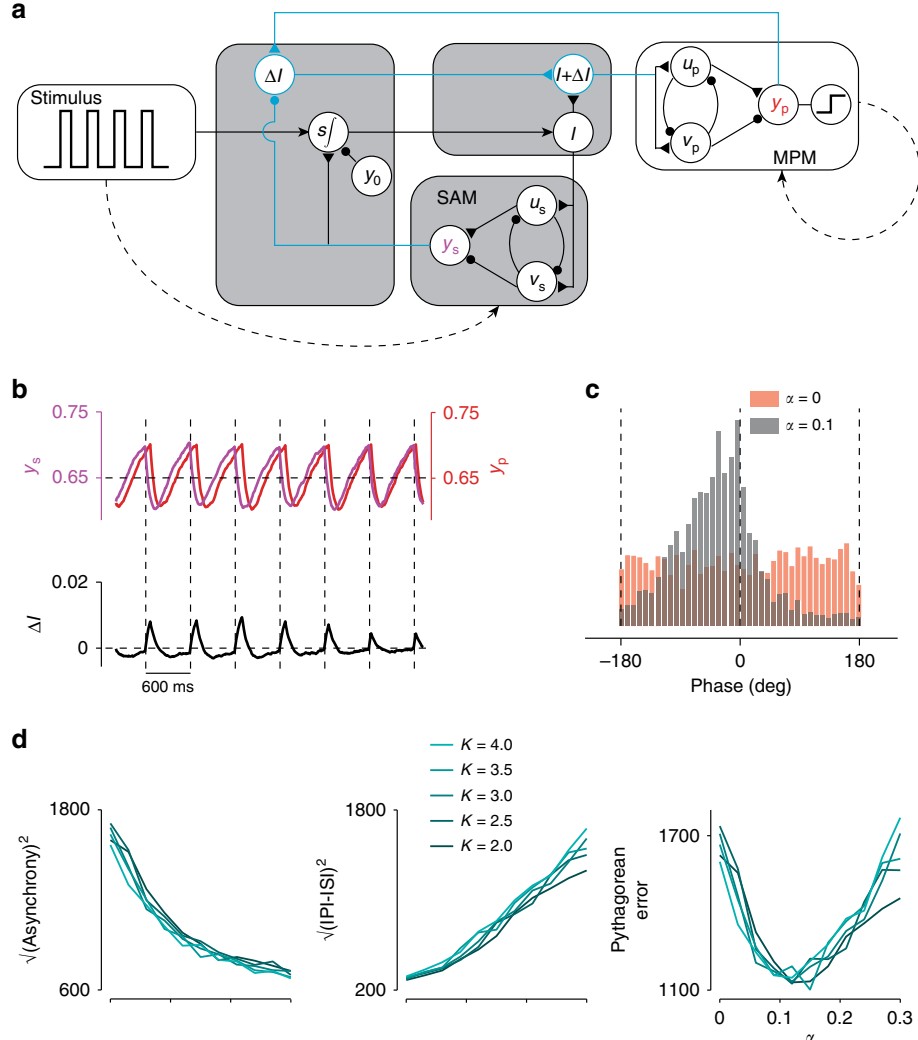

**Fig. 5 Full circuit model for synchronization. a** Augmented circuit model. To synchronize the MPM with SAM output, a second circuit pathway was introduced to measure the difference between these two outputs. The difference, weighted by $\alpha$, augments the input to the MPM with input $\Delta I$. **b** MPM output, $y_p$, SAM output, $y_s$, and their difference weighted by $\alpha = 0.1$ (i.e. $\Delta I = \alpha(y_p - y_s)$). Augmenting $I$ with $\Delta I$ adjusts controller output to the MPM such that the time of production tends to match the time of flashes (vertical dashed lines). **c** Distribution of the phase of production for the circuit with (black) and without (red) augmented input. **d** Optimization of $\alpha$. Increasing $\alpha$ decreases asynchrony (left; as defined in the Methods). In contrast, increasing $\alpha$ increases the sum of the squared IPI errors (middle). As a result, a limited range of $\alpha$ values minimize Pythagorean errors, defined as

$$\sqrt{(\mathrm{IPI} - \mathrm{ISI})^2 + (\mathrm{Asynchrony})^2} \text{ (right).}$$

synchronization tasks during which the rhythmic input was perturbed in one of three different ways: a step change in ISI (Fig. 6a, top), a phase shift (Fig. 6b, top), and a jitter in the timing of a single event (Fig. 6c, top).

When facing a sudden change in ISI (Fig. 6a, top), response times have to be adjusted in both phase and frequency. For example, after an uncued increase in ISI, the first motor response would lead the stimulus. To regain synchrony, one has to delay the response to cancel the phase lead and additionally increase the IPI to match the new ISI. Human subjects concurrently reduce asynchrony and adjust IPI such that after a few samples, actions are in sync with sensory inputs[30,32–34]. A hallmark of this error-correcting strategy is that IPIs exhibit a transient overshoot relative to the new ISI (Fig. 6a, middle)[35–37].

To test the behavior of the model in response to a change in ISI, we used the following protocol: we allowed the model to reach steady-state for an ISI of 800 ms, stepped the ISI to 1000 ms, and measured subsequent IPIs produced by the model. We

repeated this procedure 1000 times and measured how the average IPI changed over time after the step change. Qualitatively, the model exhibits the transient IPI overshoot relative to ISI that is observed in humans, and gradually adjusts the IPI to match the new ISI (Fig. 6a, bottom). Quantitatively, the degree of overshoot depends on the model's two learning rates, $K$ for frequency (Fig. 6a, bottom), and $\alpha$ for phase (Supplementary Fig. 7).

Next, we analyzed the behavior of the model in response to a phase shift, in which a single ISI is increased or decreased, causing all subsequent stimuli to occur at a different phase without any change to the subsequent ISIs (Fig. 6b, top). In this case, humans adjust their response times so that the initial mismatch is gradually reduced (Fig. 6b, middle)[38]. This model was able to capture this response pattern, and the speed of recovery could be adjusted by $\alpha$ (Fig. 6b, bottom).

Finally, we considered a perturbation in which a single stimulus is jittered temporally. This perturbation alters two consecutive ISIs in equal and opposite directions without

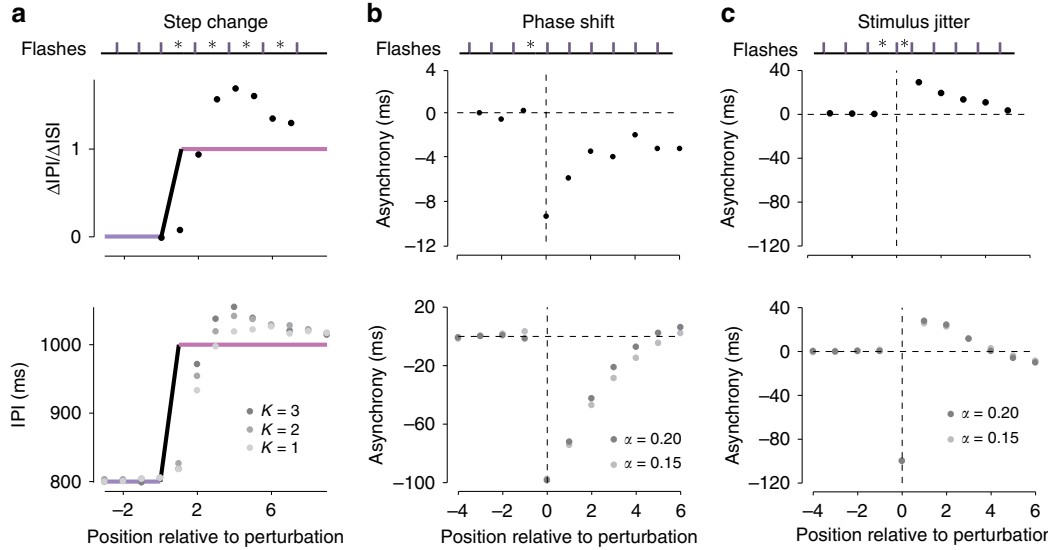

**Fig. 6 Circuit response to ISI perturbations. a** Model and human responses to a step change in the inter-stimulus-interval (ISI). Top: schematic of the task design. Vertical bars indicate the timing of a flash sequence with the distance between bars corresponding to the ISI. Asterisks denote ISIs that are perturbed relative to the initial ISI. Middle: average human step change response. Data reproduced from Bavassi et al.[37]. Circles plot the change in inter-production-interval ($\Delta$ IPI) relative to the change in ISI ($\Delta$ ISI). Blue and purple lines plot ideal performance before and after the step change, respectively. $\Delta IPI/\Delta ISI$ is defined as 0 for $\Delta ISI = 0$. Bottom: average IPI of the circuit model (circles) to a step change from 800 ms (blue line) to 1000 ms (pink line). The level of gray indicates model behavior for different values of the parameter, $K$. **b** Model and human responses to a phase shift in the stimulus timing. Top: task schematic, as in panel **a**. A phase shift was induced by increasing the duration of one interval (asterisk). Middle: average human asynchrony following a 10 ms phase shift from a 500 ms ISI. Data reproduced from Repp[38]. Bottom: average model asynchrony (circles) following a 100 ms phase shift from an ISI of 500 ms. Following the analysis of human behavior, we removed the mean asynchrony adopted by the model before the perturbation. The horizontal dashed line indicates no asynchrony and the vertical dashed line indicates the time of the first flash following the perturbation. The level of gray indicates circuit model behavior for different levels of the parameter, $\alpha$. **c** Model and human responses to temporal jitter of a single stimulus. Top: task schematic, as in panel a. The timing of a single stimulus was jittered by increasing one interval by 100 ms, followed by decreasing the subsequent interval by 100 ms (asterisks) from an ISI of 500 ms. Middle: average human asynchrony following stimulus jitter. Data reproduced from Repp[67] (asynchrony to the perturbed stimulus not shown). Bottom: average model asynchrony. Conventions as in panel (**b**).

changing either the phase or the ISI of the subsequent stimuli (Fig. 6c, top). In response to the perturbation, human subjects exhibit a characteristic change in IPI relative to ISI. For example, when a single stimulus is delayed, subjects detect the error and delay their next response accordingly. However, since the perturbation is transient, subjects have to then undo their corrective response, which is done gradually over the course of the subsequent stimuli[39,40] (Fig. 6c, middle). Again, the model was able to capture this response pattern, and the dynamics of the error correction was moderately dependent on $\alpha$ (Fig. 6c, bottom, circles).

These results demonstrate that the circuit model can capture key qualitative aspects of human behavior in response to various perturbations of stimulus timing. Behavioral studies have found individual differences in how subjects respond to these perturbations[35–40]. Accordingly, we verified that our model can capture this behavioral diversity through adjustments of model parameters ($K$ and $\alpha$) with an accuracy that is comparable to previously proposed algorithmic models[30,34,36] (Supplementary Fig. 8).

**Circuit model implements Bayesian interval reproduction.** In the absence of any prior knowledge about the ISI of the first few beats, there is no way for the circuit (or a human) to choose an informed initial speed, and only sensory feedback can guide motor timing. However, when an observer has some prior expectation about the possible values of ISIs, the optimal strategy, as prescribed by Bayesian integration, is to integrate sensory inputs with that prior knowledge. This integration enables the

system to reduce variability due to internal noise[25,41] and to estimate the ISI with greater precision.

This Bayesian integration strategy is a hallmark of human behavior in time interval reproduction experiments[18,42]. In these experiments, subjects are typically provided with a sample interval, $t_s$, drawn from a fixed prior distribution, and are asked to produce a matching interval, $t_p$. The behavior of an optimal Bayesian observer performing this task exhibits two characteristic features. First, $t_p$ values are biased toward the mean of the $t_s$ distribution. Second, the magnitude of biases becomes smaller when measurements of $t_s$ are more reliable.

Recently, we tested humans in a time interval reproduction experiment in which $t_s$ was sampled from a fixed discrete uniform distribution ranging between 600 and 1000 ms[31]. The experiment involved two trial types. In one type (Fig. 7a, left), which we refer to as "1-2-Go," subjects were presented with the first two beats of an isochronous rhythm (1-2) and were asked to produce the third omitted beat (Go). In the second type (Fig. 7a, right), referred to as "1-2-3-Go," subjects were presented with the first three beats (1-2-3) and had to synchronize their response with the fourth omitted beat. Results provided clear evidence that subjects relied on their prior knowledge, since $t_p$ values were biased toward the mean of the $t_s$ distribution (Fig. 7b, purple). Moreover, the presentation of two ISIs in the 1-2-3-Go compared to one ISI in the 1-2-Go condition allowed subjects to measure $t_s$ more accurately and reduce the bias (Supplementary Fig. 9).

We simulated the model to test if it can emulate these characteristics. The SAM received input pulses representing the beats of an isochronous rhythm (2 pulses for the 1-2-Go task and 3 for the 1-2-3-Go task) with the ISI sampled from the same

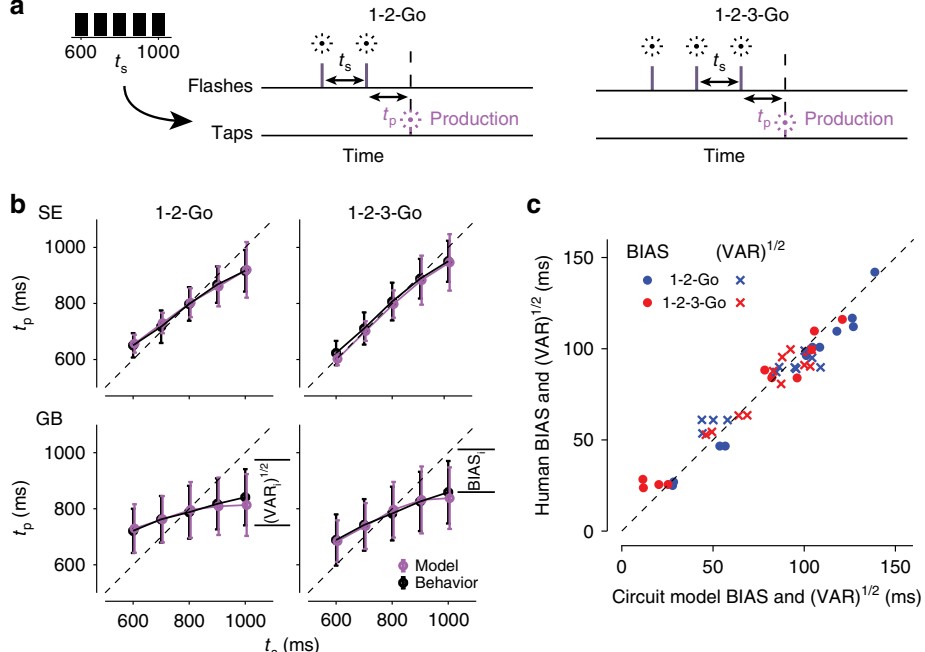

**Fig. 7 Bayesian behavior of the synchronization model. a** Schematic diagram of the interval reproduction task. In each trial, subjects were asked to produce an interval ($t_p$) that matched an interval sampled ($t_s$) from a set prior distribution (left). In 1-2-Go trials (middle), subjects observed two flashes (vertical magenta lines) demarcating $t_s$. In 1-2-3-Go trials, subjects observed three flashes (vertical magenta lines) demarcating $t_s$ twice. **b** Behavior of two example subjects (black) on the 1-2-Go and 1-2-3-Go tasks, together with the circuit fit to each subject (magenta). Error bars represent standard deviation across trials, $n = 100$ simulated trials for each $t_s$. Behavioral data modified from Egger and Jazayeri[31]. **c** Observed BIAS and VAR of nine subjects compared to the BIAS and VAR of the circuit model fit to each subject. Circles indicate BIAS, crosses VAR, and the color indicates the trial type. Dashed line represents unity.

distribution that was used in the human experiment. We defined the production interval $t_p$ of the model as the interval between the final input pulse and the next time the SAM's output exceeded $y_0$.

The behavior of the model in this task depends on three key parameters: the initial input, $I_0$, the update constant, $K$, and the level of noise in the model, $\sigma_n$. $I_0$ determines the speed of dynamics before the first measurement of $t_s$. This reflects the model's initial guess about $t_s$ based on prior expectations. $K$ determines the strength with which the SAM updates the input based on the difference between anticipated and observed $t_s$. Finally, $\sigma_n$ determines the variability of the model's behavior due to internal noise. Here, we assume that the statistics of the environment are stationary (i.e., the prior distribution is fixed), an assumption consistent with the experimental design[31]. Under these conditions, a single set of parameters is required to optimize the performance of the model. Note that we do not explicitly model the learning process that optimizes model parameters for a specific prior distribution.

Together, the parameters $I_0$, $K$, and $\sigma_n$ allow the model to capture a range of behaviors observed in human subjects (see Supplementary Table 1). Results in Fig. 7b show the behavior of the model that was fit to the data from two human subjects in the 1-2-Go and 1-2-3-Go tasks (for parameter values, see Supplementary Table 1). Evidently, the model captured several key features of Bayesian integration present in the human behavior (Fig. 7b, magenta): (1) Average $t_p$ increased monotonically with $t_s$ (Fig. 7b, black circles), (2) noise in the model caused $t_p$ to vary on a trial-by-trial basis for the same $t_s$ (Fig. 7b, black error bars), (3) average $t_p$ was biased toward the mean of the prior distribution (Fig. 7b, deviation from the dashed unity line), and (4) the magnitude of bias was smaller in the 1-2-3-Go compared to the 1-2-Go trials ($z = 2.7$, $p < 0.01$, one-sided Wilcoxon signed rank test; Supplementary Fig. 9). An alternative implementation of this

task using the full circuit architecture yields similar production times (Supplementary Fig. 10).

To further compare the model's behavior to that of the human subjects, we partitioned the root-mean-square error (RMSE) between $t_p$ and $t_s$ to two terms, a BIAS term that measures the overall bias, and a VAR term that measures average variability across all values of $t_s$ (see "Methods"). The model was able to capture the observed BIAS and VAR across subjects in both 1-2-Go and 1-2-3-Go (Fig. 7c), indicating that it can correctly implement the bias-variance trade-off exhibited by humans during interval reproduction. In this model, the update constant, $K$, plays a central role in determining the bias-variance trade-off (compare Fig. 7b, top and bottom). For a Bayesian observer, the magnitude of bias is determined by noise in the measurements of $t_s$ and by the imposed cost function. Accordingly, the value of $K$ in the model was adjusted to achieve a level of bias-variance trade-off that is inversely related with the inherent noise in the model ($\sigma_n$) and the operative cost function ($r = -0.94$, $p = 0.0002$; see Supplementary Table 1). These results provide a potential mechanistic account of linear updating algorithms proposed previously to describe Bayesian timing behaviors[43] (Supplementary Fig. 11).

**Bayesian synchronization/continuation.** As our final test, we examined the model in a Bayesian synchronization/continuation task that demands sensory anticipation, motor timing, and Bayesian integration. In this task, subjects are asked to tap synchronously with a metronome (synchronization phase) and continue to tap with the same tempo without the metronome until instructed to stop (continuation phase). In the classic version of this task, subjects have no uncertainty about the ISI of the metronome as it is kept constant across trials. Recently, humans were tested on a variant of this task in which ISI for each trial was

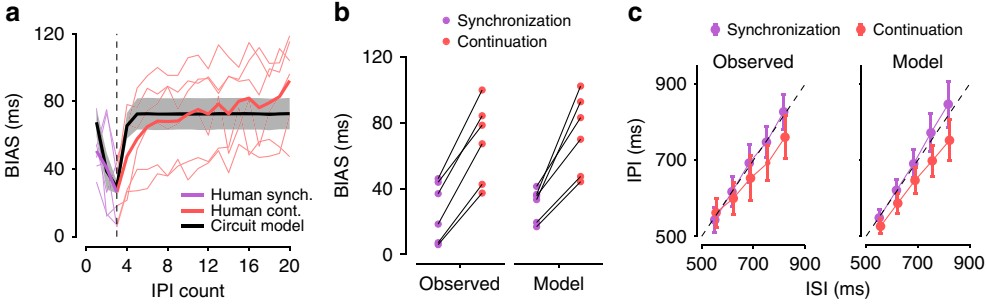

**Fig. 8 Circuit model captures systematic biases in human behavior during Bayesian synchronization/continuation. a** Overall BIAS in the synchronization/continuation task. BIAS was calculated as in Fig. 7. Purple and red lines indicate BIAS for human subjects during synchronization and continuation, respectively. Thin lines indicate individual subjects. Thick lines indicate the mean across subjects. The black line with shading indicates the mean and standard deviation of the BIAS in the circuit model fit to individual subjects. The vertical dashed line marks the transition from synchronization to continuation. **b** Observed and model BIAS for individual subjects during each phase of the task. Lines connect data points for each subject or their respective model fits. **c** Inter-production-interval (IPI) for different inter-stimulus-intervals (ISIs) for an example subject (left) and the circuit model fit to that subject's behavior (right). Dashed lines indicate unity. Behavioral data for all panels modified from Narain et al.[44]. Error bars represent standard deviation across trials, $n = 21$ simulation trials for each ISI.

sampled randomly from a prior distribution[44]. While performing this task, humans exhibit distinct patterns of biases in their IPIs. During synchronization, responses were biased toward the mean of the ISI distribution and the magnitude of the bias decreased with the number of synchronization stimuli (Fig. 8a, purple lines). These observations are consistent with the Bayesian integration scheme that we discussed previously in the context of a time interval reproduction task (Fig. 7). During the continuation phase, the bias persisted and its magnitude increased substantially (Fig. 8a, red lines) for all subjects (Fig. 8b). This increase in bias was due to a drop in the sensitivity of IPI to ISI (shallower slope relating IPI to ISI) as well as an overall bias toward shorter IPIs (Fig. 8c, left).

We fitted the model parameters ($I_0$, $K$, and $\alpha$) to individual subject's behavior in this task (see "Methods"; Supplementary Table 2). Model fits generated IPIs that were biased towards the mean ISI. Consistent with subjects' behavior, the magnitude of the bias decreased during synchronization and increased during continuation (Fig. 8a, black line). Moreover, like humans, the increase in bias during continuation was due to a combination of decreased sensitivity to the ISI (decrease in slope of IPI-ISI relationship, $z = 2.1$, $p = 0.02$, one-sided Wilcoxon signed rank test) and an overall shift towards shorter IPIs (decrease in IPI of the middle interval, $z = 2.1$, $p = 0.02$, one-sided Wilcoxon signed rank test; Fig. 8c, right; Supplementary Fig. 12).

In the model, this pattern of biases is explained by the augmented input, $\Delta I$, to the MPM, which is necessary for the full model to synchronize its outputs with sensory inputs. Following the final stimulus input in the synchronization phase, the SAM no longer resets and instead proceeds toward the terminal fixed point. As a result, its output, $y_s$, becomes fixed at a value greater than $y_0$. However, $y_p$ continues to oscillate between values smaller than $y_0$ because it is reset by the motor output. This mismatch makes $\Delta I$ negative and decreases the overall input to the MPM. The decreased input, in turn, speeds up the dynamics of the MPM and shortens IPIs during the continuation phase across all ISIs. Further, the impact of this negative $\Delta I$ is integrated over time, leading to an asymmetric impact on the speed of MPM dynamics associated with a short ISI compared to a long ISI. Specifically, longer ISIs lead to a greater increase in speed than shorter ISIs, accounting for the decreased sensitivity to the ISI during continuation. It is important to note that these results require that the SAM does not reset. Therefore, an alternate circuit configuration in which the output of the MPM resets the SAM

will not be able to capture the increase in bias during continuation. This finding validates our assumption that resetting the SAM relies on sensory inputs and not motor outputs. Further, this feature of the circuit model allowed it to more accurately capture the observed pattern of bias than simple linear updating algorithms that have previously been used to model human synchronization behavior (Supplementary Fig. 13).

## Discussion

To coordinate movements with external stimuli, the dynamics of neural activity must be adjusted based on sensory inputs[7,15,45]. However, precise control of dynamics is challenging because neural signals are subject to internal noise, and timing cues are often discrete and delayed[21]. Here, we proposed a simple neural circuit model that can address these challenges in a wide range of timing tasks.

The model is comprised of two modules, an MPM that generates timed actions, and a SAM that predicts the time of upcoming external events. Both modules were engineered to emulate a recently discovered speed control mechanism in the frontal cortex of monkeys during a flexible motor timing task[7]. The MPM uses this speed control mechanism in conjunction with a reset to produce rhythmic outputs. The SAM uses the same speed control mechanism in conjunction with an error-correction scheme that enables it to predict the timing of external events.

Coupling the MPM and SAM creates a closed-loop control system with versatile timing capacities in the presence of noise and delayed sensory feedback. For example, the model was able to capture human behavior in classic beat synchronization tasks[30,32–34,37]. In this case, the function of the SAM was to detect and correct discrepancies between the model's output and external beats so that the MPM could adjust the outputs accordingly. The model was also able to emulate the biases associated with Bayesian time interval reproduction in the presence of noisy measurements when sample intervals were drawn from a fixed prior distribution[18,31]. To do so, the initial input to the model had to be adjusted based on the prior distribution, and the model's updating parameters had to take the magnitude of the noise into account. Finally, a phase-correction mechanism between the SAM and MPM enabled the model to capture several non-trivial features of human behavior during a Bayesian synchronization/continuation task[44] that previous algorithmic models could not.

Previous studies have proposed algorithmic models for how humans coordinate their motor outputs with sensory inputs. The basic algorithm is to compare the time of each motor output to the time of the corresponding sensory input, and use the error between the two to adjust the timing of the next movement[31–34,43]. This algorithm is conceptually similar to how our model functions, and its error-correcting behavior in beat synchronization tasks is comparable to our model[30,43] (Supplementary Fig. 8). However, by virtue of implementing this algorithm using a neural circuit, our model's predictions can be readily compared to neural data. Moreover, our model provides a mechanistic understanding of hypothesized predictive processes that algorithmic models have attributed to human behavior during perceptual timing tasks[43,46,47].

Previous circuit models of timing largely fall into three classes, each with their own strengths and weaknesses. The first class is based on the accumulation of ticks of a central clock[11,48–50]. Similar to our circuit model, these clock-accumulator models rely on ramping activity that is observed in individual neurons[12,13]. However, these models do not explain how the recurrent circuit interactions lead to such ramping activity. The second class uses large recurrent neural circuits capable of producing rich dynamics[2,7,51–53]. These models can produce activity patterns that are strikingly similar to those observed in local populations of neurons, but it is unclear how they can flexibly integrate sensory and motor feedback. The third class uses a system of coupled oscillatory units[26–28,54,55]. These models can produce sophisticated timing behaviors and can integrate sensory information across a range of time scales[26]. However, the activity profile of neurons in the brain regions causally involved in timing is typically not oscillatory[7]. Our model provides an understanding of the link between these model classes. First, it explains the ramping activity in terms of recurrent dynamics due to interactions between neurons. Second, the model's dynamics can be flexibly adjusted based on incoming sensory input and motor feedback. Third, the model generates oscillations at its output. However, unlike abstract models comprised of inherently oscillatory units, our model generates oscillations through circuit-level interactions that are consistent with the speed control mechanism observed in single neurons in the frontal cortex[7].

Finally, the wiring of our circuit model might provide insight into the individual functional contributions of the cortical and subcortical systems important to action and perceptual timing[56]. One node, the basal ganglia, has been heavily implicated in action timing through lesion studies[57] and physiological evidence[7,45,58]. An important principle of the basal ganglia function is the inhibition of downstream neural activity[59]. Given their proposed function as a competitive selection mechanism[60], these inhibitory pathways may be the substrate for implementing the mutual inhibitory interactions needed for the temporal control of movements[6]. Another key node, the cerebellum, has also been linked to timing through lesion studies[61], physiology[62,63], and modeling[44]. A hallmark of cerebellar function is the detection and correction of sensory errors during sensorimotor control[64,65], which is the key function of the SAM in our model. Finally, the output of both the basal ganglia and cerebellum are sent transthalamically to regions of the frontal cortex involved in sensory and motor timing[66]. Accordingly, we speculate that the integration of the sensory anticipation and motor production modules in our model may rely on interaction between the basal ganglia, cerebellum, and frontal cortical areas[66].

## Methods
Modeling and analyses were performed in Python 3.5.4 and Matlab R2017a.

**BCM for interval production.** The fundamental circuit architecture consists of three rate units, $u$, $v$, and $y$, which are governed by the following set of equations:

$$\tau \frac{du}{dt} = -u + \theta\big(W_{uI}I - W_{uv}v + \eta_u\big), \tag{8}$$

$$\tau \frac{dv}{dt} = -v + \theta\big(W_{vI}I - W_{vu}u + \eta_v\big), \tag{9}$$

$$\tau \frac{dy}{dt} = -y + W_{yu}u - W_{yv}v + \eta_y, \tag{10}$$

where $\tau$, the time constant of each unit, was set to 100 ms and $\theta(x)$, the activation function of each unit, was specified by $\theta(x) = 1/[1 + \exp(-x)]$, and $I$ is a tonic input. $W_{uI}$, $W_{uv}$, $W_{vu}$, and $W_{vI}$ specify the weighting of the interactions between units and each was set to 6 for all simulations. Similarly, we set $W_{yu} = W_{yv} = 1$. Noise in the system was modeled by 0 mean Gaussian white noise inputs, represented by the variables $\eta_u$, $\eta_v$, and $\eta_y$ for unit $u$, $v$, and $y$, respectively. These variables were independently sampled at every time point from a Gaussian distribution with mean set to 0 and standard deviation specified by $\sigma_n$. The units were initialized at $u = 0.7$, $v = 0.2$, $y = 0.5$, and $I = I_0$, where $I_0$ is a free parameter. All simulations were carried out using Euler's method with a step size of 10 ms.

**Motor planning module.** The MPM consists of four rate units $I$, $u_p$, $v_p$, and $y_p$, which are simulated based on the following equations:

$$\tau \frac{du_p}{dt} = -u_p + \theta\big(W_{u_pI}I - W_{u_pv_p}v_p + \eta_{u_p} - I_p\big), \tag{11}$$

$$\tau \frac{dv_p}{dt} = -v_p + \theta\big(W_{v_pI}I - W_{v_pu_p}u_p + \eta_{v_p} + I_p\big), \tag{12}$$

$$\tau \frac{dy_p}{dt} = -y_p + W_{yu_p}u_p - W_{yv_p}v_p + \eta_{y_p}. \tag{13}$$

$I_p$ specifies a transient input to $u_p$ and $v_p$ that serves to reset the system when $y_p > y_0$. We set $I_p$ to 50 and $y_0$ to 0.7 for all simulations. The activity level of $I$ controls the speed of the dynamics of the MPM. The units were initialized at $u_p = 0.7$, $v_p = 0.2$, $y_p = 0.5$. Noise, represented by the variables $\eta_{u_p}$, $\eta_{v_p}$, and $\eta_{y_p}$, was injected into each unit as in the BCM. All other parameters are the same as the BCM.

**Sensory anticipation module.** The SAM consists of four rate units $I$, $u_s$, $v_s$, and $y_s$. The system evolves according to the equations

$$\tau \frac{du_s}{dt} = -u_s + \theta\big(W_{u_sI}I - W_{u_sv_s}v_s + \eta_{u_s} - sI_s\big), \tag{14}$$

$$\tau \frac{dv_s}{dt} = -v_s + \theta\big(W_{v_sI}I - W_{v_su_s}u_s + \eta_{v_s} + sI_s\big), \tag{15}$$

$$\tau \frac{dy_s}{dt} = -y_s + W_{y_su_s}u_s - W_{y_sv_s}v_s + \eta_{y_s}, \tag{16}$$

$$\tau \frac{dI}{dt} = sK\big(y_s - y_0\big), \tag{17}$$

where $K$ is a free parameter and $\eta$ indicates noise input to each unit as in the BCM. The module receives a binary input $sI_s$, where $s$ represents the visual stimulus ($s = 1$ when the stimulus is on and $s = 0$ when the stimulus is off) and $I_s$ is set to 50. This input serves to reset the values of $u_s$ and $v_s$ at the time of each stimulus. All other parameters are the same as the BCM.

Sequences of stimuli were implemented by setting $s = 1$ periodically. Each stimulus presentation lasts 10 ms. The units $u_s$, $v_s$, and $y_s$ were initialized as in the MPM and $I$ was initialized at $I_0$, a free parameter. The dynamics were allowed to run forward for 750 ms before the first stimulus was presented. This initialization period serves to put the circuit closer to steady-state so that the effect of reset signals is consistent across different ISIs. $I$ does not update during the first presentation of the stimulus (by setting $K = 0$ during the first stimulus presentation).

Optimization of SAM parameters was achieved by simulating the module with 100 different pairs of $I_0$ and $K$ for each $\sigma_n$. $K$ was uniformly sampled from values between 1 and 8.0, and $I_0$ was sampled from values between 0.77 and 0.79. $N$ stimuli were presented to the model, where $N = 2, 3, \dots, 10$. ISIs were sampled from a discrete uniform distribution with five values between 600 and 1000 ms.

For each set $(\sigma_n, I_0, K, N)$, the model was run as described above and the RMSE was calculated according to

$$\text{RMSE} = \sqrt{\frac{1}{N_s} \sum_{n=1}^{N_s} (\text{IPI}_n - \text{ISI}_n)^2}, \tag{18}$$

where $N_s = 500$, the total number of iterations across all ISIs. The pair $(I_0, K)$ that

results in the smallest RMSE was recorded. The procedure was repeated 10 times for each $\sigma_n$ to obtain a distribution of the optimal $(I_0, K)$.

In Fig. 3c, $I_0$ and $K$ were picked from a $30 \times 30$ grid spanning the the same range as the optimization procedure, and $N_s = 5000$.

**Behavioral tasks**. We used a suite of behavioral tasks to test circuit model timing performance. In all tasks, we performed numerical simulations of the dynamical equations expressed above using Euler's method and a time step of 10 ms. All behavioral experiments were performed with the approval of the Committee on the Use of Humans as Experimental Subjects at MIT after receiving informed consent.

**Periodic production task**. To achieve periodic production, we simulated the MPM in isolation at four different levels of input, $I$, uniformly spaced from 0.75 to 0.78, for 40 s at each level. To measure the performance of the model, we calculated the IPI as

$$\text{IPI}_n = t_{n+1} - t_n, \tag{19}$$

where $t_n$ was the time of the $n$th action produced by the model. For all levels of input, we set $\sigma_n = 0.01$. To compare the timing behavior of the model across levels of input, we calculated the mean and standard deviation of the first 40 IPIs generated by the circuit.

**ISI tracking task**. Circuit model synchronization was tested in a randomized ISI tracking task. The ISI was defined as

$$\text{ISI}_n = m_{n+1} - m_n, \tag{20}$$

where $m_n$ was the time of the $n$th stimulus. On each trial of the task, we initialized the ISI at 800 ms and simulated a sensory input with 20 consecutive 800 ms intervals. After the first block of 20 ISIs, the ISI for the next 20 intervals was selected at random from a discrete uniform distribution between 600 and 900 ms with four possible values. This process was repeated four times, to generate five blocks of ISIs, corresponding to a sequence of 100 total intervals.

Circuit parameters were selected as follows: $I_0$ was set to 0.771; $K$ was selected to be 2; $\alpha$ was chosen to be 0 for simulations with no augmented input (e.g. Fig. 4) and 0.1 for simulations with augmented input (e.g. Fig. 5). $I_0$ was set such that the MPM generated an 800 ms IPI on average, matching the initial ISI of every trial. This ensured that differences in circuit output resulted from updates to the value of $I$, rather than the value of $I_0$ selected.

The $n$th IPI was quantified as in Eq. (19) and compared to the $n$th ISI. The asynchrony between an action and the stimulus presented at time $m_n$ was defined as

$$a_n = t_n - m_n, \tag{21}$$

where $t_n$ was the time of the action which was closest in time to $m_n$. The relative phase of the $n$th action, $\phi_n$, was calculated as

$$\phi_n = 360 \frac{a_n}{\text{ISI}_n}, \tag{22}$$

where the phase is in units of degrees. For all simulations, we set $\sigma_n = 0.01$.

**ISI perturbation task**. The basis of the mechanisms behind human synchronization behavior are generally probed using perturbations of the ISI after an experimental subject has reached steady-state synchronization performance to a given ISI (see Repp[30] for a review). To compare the circuit model synchronization performance to that of humans, we explored the behavior of the model in response to three common ISI perturbations: (1) a step change in the ISI, where the circuit synchronized to a stimulus with an ISI of 800 ms before the ISI was stepped to 1000 ms for all subsequent ISI. (2) A phase shift in the stimulus timing, where the circuit synchronized to a stimulus with a 500 ms ISI before the timing of stimuli was shifted by increasing a single ISI to 600 ms. All subsequent ISIs were 500 ms. (3) Stimulus jitter, where the circuit synchronized to an ISI of 500 ms before the timing of a single stimulus is perturbed, while subsequent stimuli remained in phase with the stimulus before the perturbation. To accomplish this, we perturbed two successive ISIs; the first was increased to 600 ms and second was decreased to 400 ms.

We simulated 1000 trials of each perturbation type and calculated the asynchrony associated with each action as in Eq. (21) and the mean IPI between actions as in Eq. (19). To ensure the circuit model was fully synchronized before a perturbation, we simulated 30 ISIs of the same duration before applying the perturbation. The circuit model was simulated with $I_0 = 0.771$ and $\sigma_n = 0.005$. The values of $K$ and $\alpha$ were varied to explore the behavior of the model with different sensitivities to errors in simulation and synchronization.

**1-2-Go and 1-2-3-Go tasks**. We compared the circuit model interval reproduction behavior to that of humans performing a timing task we refer to as 1-2-Go and 1-2-3-Go. The methods used for testing human interval reproduction are summarized briefly here. Please see our behavioral paper for a full description of the task and methods[31].

The behavior of nine human subjects was analyzed. Subjects sat in a dark, quiet room at a distance of approximately 50 cm from a display monitor. The display monitor had a refresh rate of 60 Hz, a resolution of 1920 by 1200, and was

controlled by a custom software (MWorks; http://mworks-project.org/) on an Apple Macintosh platform.

The interval reproduction task consisted of two randomly interleaved trial types "1-2-Go" and "1-2-3-Go". On 1-2-Go trials, two flashes demarcated a sample interval ($t_s$). On 1-2-3-Go trials, three flashes demarcated $t_s$ twice. For both trial types, subjects had to reproduce $t_s$ immediately after the last flash by pressing a button on a standard Apple keyboard. On all trials, subjects had to initiate their response proactively and without any additional cue. Subjects received graded feedback on their accuracy.

Each trial began with the presentation of a 0.5° circular fixation point at the center of a monitor display. The color of the fixation cued the trial type. After a 2 second delay, a warning stimulus and a trial cue were presented. The warning stimulus was a white circle that subtended 1.5° and was presented 10° to the left of the fixation point. The trial cue consisted of 2 or 3 small rectangles 0.6° above the fixation point (subtending 0.2° × 0.4°, 0.5° apart) for the 1-2-Go and 1-2-3-Go trials, respectively. After a random delay with a uniform hazard (250 ms minimum plus an interval drawn from an exponential distribution with mean of 500 ms), flashes demarcating $t_s$ were presented. Each flash lasted 6 frames (~100 ms) and was presented as an annulus around the fixation point with an inside and outside diameter of 2.5° and 3°, respectively. $t_s$ was sampled from a discrete uniform distribution ranging between 600 and 1000 ms with a 5 samples. To help subjects track the progression of events throughout the trial, after each flash, one rectangle from the trial cue would disappear (starting from the rightmost). The produced interval ($t_p$) was measured as the interval between the time of the last flash and the time when the subject pressed the response key. We discarded any trial when the subject responded before the second (for 1-2-Go) or third (for 1-2-3-Go) flash and any trial where the response was 1000 ms after the veridical $t_s$. We further used a model based approach to identify "lapse trials," or trials in which $t_p$ was not related to $t_s$[31]. We then calculated the mean $t_p$ conditioned on $t_s$ and trial type.

The SAM was run for 100 simulated trials for each $t_s$ (600 ms, 700 ms, 800 ms 900 ms, 1000 ms). To simulate the 1-2-Go task, the SAM was presented with 2 stimuli that were separated by the selected $t_s$. The 1-2-3-Go task was simulated similarly but with 3 stimuli. The production interval $t_p$ of the model was defined as the interval between the final input pulse and the next time $y_s$ exceeded $y_0$.

For each subject, we calculated the mean and standard deviation of $t_p$ for each $t_s$ in both the 1-2-Go and 1-2-3-Go tasks (10 task conditions in total). We call these values $\mu_{1,\text{subject}}$, $\text{STD}_{1,\text{subject}}$, . . . , $\mu_{10,\text{subject}}$, $\text{STD}_{10,\text{subject}}$ (one pair of values for each condition).

For each set of parameters ($\sigma_n$, $I_0$, $K$), the model was run as described above. We calculated the mean and standard deviation of the model's $t_p$ for each $t_s$, and for both the 1-2-Go and 1-2-3-Go tasks. We call these values $\mu_{1,\text{model}}$, $\text{STD}_{1,}$ $_{\text{model}}$ . . . , $\mu_{10,\text{model}}$, $\text{STD}_{10,\text{model}}$.

The fitting was done by alternating between fitting $\sigma_n$ and jointly fitting ($I_0$, $K$). The parameters were uniformly and independently sampled from the following intervals: [1.0, 8.0] for $K$, [0.77, 0.79] for $I_0$ and [0.005, 0.4] for $\sigma_n$. For each step, 100 sets of parameters were sampled and the optimal set was used to update the parameters.

$$\sigma_n = \operatorname*{argmin}_{\sigma_n} \left( \sum_{k=1}^{10} \left( \text{STD}_{k,\text{model}} - \text{STD}_{k,\text{subject}} \right)^2 \right), \tag{23}$$

$$I, K_0 = \operatorname*{argmin}_{(I,K_0)} \left( \sum_{k=1}^{10} \left( \mu_{k,\text{model}} - \mu_{k,\text{subject}} \right)^2 \right). \tag{24}$$

To evaluate the anticipated $t_s$ associated with $I_0$ and $\sigma_n$, we initialized the model with $I_0$, presented a single stimulus to the model 750 ms after initialization, and determined the interval between this stimulus and the next time $y_s$ exceeded $y_0$.

We defined BIAS and VAR for each subject and model simulation as

$$\text{BIAS}^2 = \frac{1}{N} \sum_{i=1}^{N} \left( \bar{t}_{p_i} - t_{s_i} \right)^2, \tag{25}$$

$$\text{VAR} = \frac{1}{N} \sum_{i=1}^{N} \sigma_i^2, \tag{26}$$

where $N = 5$ is the number of target intervals, $\bar{t}_{p_i}$ and $\sigma_i^2$ are the mean and variance of production intervals that correspond to the target interval $t_{s_i}$.

**Synchronization/continuation task**. We compared the circuit model to the behavior of humans performing a synchronization/continuation task in which the inter-stimulus-interval (ISI) was selected at random each trial from a set distribution. The methods used for testing human experiments are summarized briefly here. Please see Narain et al. [44] for a full description of the task and methods[44].

We analyzed the behavior of six human participants. Stimuli were viewed from a distance of approximately 67 cm in a dark, quiet room. The display monitor had a refresh rate of 60 Hz, a resolution of 1920 by 1200, and was controlled by a custom software (MWorks; http://mworks-project.org/) on an Apple Macintosh platform.

Each trial began with the presentation of a red circular fixation stimulus (diameter = 0.75° visual angle) in the center of the screen. After a variable

delay (200 ms plus an additional amount of time which was exponentially distributed with mean = 300 ms and a maximum value of 2300 ms), a synchronization stimulus was flashed four times with an ISI chosen from a discrete uniform distribution (five intervals, minimum = 550 ms, maximum = 817 ms). The flashing stimulus was a gray annulus centered around fixation with an inner diameter of 1° and outer diameter of 1.25°. Participants were instructed to tap a response key synchronously with the third and fourth synchronization flashes and continue tapping to reflect the ISI until instructed to stop. The number of continuation taps required was three plus an exponentially distributed number with a mean of nine and a maximum of 22. The first IPI was defined as the interval between the middle of the second flash and the first key press. Subsequent IPIs were defined as the interval between successive key presses.

Biases in IPIs were calculated according to the following procedure. For each interval, $ISI_i$, we found the mean IPI for the $k$th interval in the sequence ($\overline{IPI}_{i,k}$). The mean squared bias for each $k$ was defined as

$$BIAS_k^2 = \sum_{i=1}^{N} \frac{1}{N} \left( \overline{IPI}_{i,k} - ISI_i \right)^2, \qquad (27)$$

where $N = 5$ is the number of distinct ISI values.

We simulated 21 trials of the full synchronization model for each ISI. In each trial, the stimuli presented to the model were a series of three flashes separated by the ISI of that trial. After the three flashes, the stimulus was set to $s = 0$ and the model was run and allowed to produce 17 more productions. The IPI of the circuit was defined as the time difference between successive productions.

We fixed $\sigma_n$ at 0.01 and varied three free parameters $K$, $I_0$, and $\alpha$. For each subject, we simulated 100 different parameter combinations with $K$ randomly selected from the interval [0.01, 5], $I_0$ from the interval [0.76, 0.78] and $\alpha$ from the interval [0.01, 0.1]. For each combination of parameters, the model was run as described above and bias was calculated as in Eq. (27). We then found the combination of parameters that minimized the mean squared errors between the $BIAS_k$ observed in the subject and the $BIAS_k$ of the circuit, across $k = 1, \ldots, 20$.

Synchronization bias was quantified as the bias for $k = 3$, and continuation bias was the bias for $k = 7$.

**Linear behavioral algorithm for sensorimotor timing**. For comparison purposes, we also developed a linear updating model for synchronization based on previous work[30]. In the absence of any error, the model computes the $(n + 1)$th production time, $t_{n+1}$, by adding the expected ISI, $T_n$, to the $n$th production time, $t_n$. When there is a discrepancy, the updating rule incorporates two additional error terms. The first error term measures the asynchrony, which is the difference between $t_n$ and the time of the $n$th metronomic input, $m_n$. The second error term measures the difference between $T_n$ and the observed ISI, $ISI_n$. In the model, these two error terms are weighted by $\beta_{Asynch}$ and $\beta_{ISI}$, respectively.

$$t_{n+1} = t_n + T_n - \beta_{Asynch}(t_n - m_n) - \beta_{ISI}(T_n - ISI_n). \qquad (28)$$

We simulated this timing model for different values of $\beta_{Asynch}$ and $\beta_{ISI}$ to compare this standard algorithm to the behavior of the circuit model.

In simulations of the 1-2-Go and 1-2-3-Go tasks, we added a noise term, $\eta$, to $t_{n+1}$. $\eta$ was sampled independently at every time point from a Gaussian distribution with mean 0 and standard deviation $\sigma_n$. For fitting, $\beta_{Asynch}$ was set to 0 and parameters were uniformly and independently sampled from the following intervals: [0.0, 1.0] for $\beta_{ISI}$, [700, 900] for $T_0$, and [30, 120] for $\sigma_n$. The fitting procedure is the same as described for the circuit model.

In simulations of the synchronization/continuation task, the production times $t_n$ during the synchronization phase were simulated like the 1-2-3-Go task. For the continuation phase, $t_{n+1} = t_n + T_N + \eta$, where $\eta$ was sampled independently at every time point from a Gaussian distribution with mean 0 and standard deviation $\sigma_n$, and $T_N$ is the value of $T$ at the final stimulus presentation. For fitting, $\sigma_n$ was set to 50, and parameters were uniformly and independently sampled from the following intervals: [600, 1200] for $T_0$, [0, 2] for $\beta_{ISI}$, [0, 1] for $\beta_{Asynch}$. The fitting procedure is the same as described for the circuit model.

**Reporting summary**. Further information on research design is available in the Nature Research Reporting Summary linked to this article.

## Data availability
No new data were collected. Data for previously published work is available at https://jazlab.org/resources/.

## Code availability
Code used in this study is publicly available at https://jazlab.org/resources/.

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

## Acknowledgements
The authors thank Jing Wang, Morteza Saraf, Nicolas Meirhaeghe, and Alexandra Ferguson for their insightful comments on this manuscript and Michal De-Medonsa for her editorial assistance. M.J. is supported by NIH (NINDS-NS078127), the Klingenstein Foundation, the Simons Foundation, the McKnight Foundation, and the McGovern Institute.

## Author contributions
S.W.E. conceived of the extension of the basic circuit module. S.W.E. developed, implemented and characterized the motor planning module. N.M.L. developed, implemented and characterized the sensory anticipation module. S.W.E. developed the circuit for the synchronization task and characterized model performance. N.M.L. compared model performance to humans in the 1-2-Go, 1-2-3-Go, and synchronization/continuation tasks. M.J. supervised the project. All authors contributed to the interpretation of results and writing of the manuscript.

## Competing interests
The authors declare no competing interests.
