## [Peer Review File · Nature Communications]

Reviewers' Comments:

Reviewer #1:

Remarks to the Author:

In the paper "A neural circuit model for human sensorimotor timing", the authors present a model that can produce temporally precise intervals that match the intervals of a stimulus. This work addresses an important outstanding problem in computational neuroscience – how do neural circuits generate temporally precise outputs and adapt to a changing world? As the authors note, this problem has been studied extensively with multiple suggested solutions (e.g., accumulators, oscillators, RNNs) though each has limitations (e.g., unclear neural implementation, inflexibility, etc.). The authors' proposed model, on the other hand, is able to account for a wide range of phenomena and could be implemented neurally.

Specifically, the authors propose a module of two mutually inhibiting units, u and v , which both receive a common excitatory input I and compete with each other via winner-take-all dynamics. The strength of I sets the time-scale of this competition. Units u and v both project to an output y , the former with an excitatory and the latter with an inhibitory connection. The interval measured by the module is given as the time when y crosses some threshold. The speed of this occurring is set by the time-scale of resolving the competition between u and v , which in turn is set by the input I . Thus, a specific I is mapped to a specific interval. After the interval, u , v , and y are reset and the dynamics are repeated.

The authors employ the same basic module twice: once to predict the timing of a stimulus that is assumed to be periodic, and the other to produce outputs. The authors then show that: (1) by using the discrepancy, at the stimulus times, between the y of the prediction module and the threshold, the input I can be updated to improve future predictions; and (2) by using the discrepancy between the y 's of the prediction and production modules, the input I can be updated to obtain phase-locking between the outputs and a periodic stimulus.

This full model can be fit to human behavioral data in a series of interval timing tasks (e.g., synchronization, ready-set-go, continuation after synchronization) and a good match to the behavior can be found. The authors can then study the parameters of the model and make inferences as to the sources of behavioral variability in human subjects.

Overall, this work contributes novel ideas to the field and merits publication. I have only two concerns about which I would like to query the authors.

1 - Mapping of parameter space. There are several parameters of the model that I

would like to understand further. The range of I used for the results is very narrow (0.75–0.78). What is the model's behavior outside of this range? I.e., is the relationship to the IPI monotonic over a larger range of I ? What is the noise level in Figs. 1a and 3b (I can't find these values in the text)? Does performance fall apart at higher noise levels? How important are the initial values of u and v and the 750ms initialization period? The authors cite Wang et al., Nature Neuroscience, 2018, for the motivation for their basic computational module. Given that this is a pure theory paper, I believe that plotting the phase space of u and v (as in Wang 2018) would be useful here.

2 - The interval reproduction tasks. If I understand correctly, only the prediction module is being used for the interval reproduction tasks, with the reproduction time defined as the threshold crossing for the prediction module. I assume this is because, in the full model, the production module requires multiple stimulus repeats to phase-lock appropriately. However, human subjects are able to produce a motor output with the correct timing after only one (or two) ISIs. I believe the authors should specifically address this issue with respect to their model. Furthermore, I believe the definition of t_p used in Fig. 7 should be stated clearly in either the main text or the figure caption.

Minor points:

- Recommend changing "Because y_p is driven by u_p and v_p , this results..." to "Because y_p is driven by u_p and inhibited by v_p , this results..." for clarity.
- Recommend adding y-axes values to Fig. 2b.
- Can you expand on why there is no updating of I at S1 for Fig. 3? It's not obvious to me why that would break the necessary adaptation.
- In Fig. 3b, you have a nice dotted line for y_0 . I recommend using a similar dotted line for I_0 .
- The quantitative definition of asynchrony (i.e., the y-axis of the leftmost plot in 5d) is not referenced in the main text or the figure caption. I recommend alerting the reader at some point that this is defined in the Methods as the authors typically do.
- In Fig. 6, I initially assumed that the flash times in the top row were aligned with the "Position relative to perturbation" x-axes below, which they are not. For clarity, I recommend these be aligned.
- In 6a, the purple, black, and pink lines are never defined or referenced in the top plot (I understand they correspond to optimal performance). By the way, I assume defining $\Delta IPI/\Delta ISI$ to be zero prior to the perturbation is a convention since $\Delta ISI=0$ so the ratio is undefined.
- In 6a, the human data show an immediate overshoot on the first trial after the perturbation is detected (black dot above the pink line) while the model takes two

trials after error detection to overshoot. Is this just because K is not large enough?

- In 6b–c, the authors note that they remove the “mean asynchrony”. Is this justified? Why is there a mean asynchrony in the model whilst not in the human data? Can the author propose a possible fix for this?

- The authors indicate that BIAS and VAR are defined in the Methods. Only the former is.

- This reviewer notes that he is on a paper relevant to the sentence “These models rely on the rich set of dynamics generated by neurons connected in a local circuit to represent the passage of time”. The paper is Depasquale et al., PLOS One, 2018 (Fig. 7 is the relevant figure).

- This reviewer notes that he is on a paper relevant to the sentence “Given their proposed function as a competitive selection mechanism, these inhibitory pathways may be the substrate for implementing the mutual inhibitory interactions needed for the temporal control of movements”. The paper is Murray and Escola, 2017 (which the authors cite elsewhere; the paper is entirely about mutual inhibition for the temporal control of the motor system).

It was a pleasure to read and review this paper. Thank you for the opportunity.

Sean Escola, M.D., Ph.D.
Assistant Professor of Psychiatry
Center for Theoretical Neuroscience
Columbia University

Reviewer #2:

Remarks to the Author:

In this manuscript authors present a model of the interaction between sensory and motor timing, and how motor timing can be entrained by rhythmic sensory stimuli. The core of the model is a pair of mutually inhibitory units that are driven by a shared input, and an output unit driven by the difference of the two recurrent units. Because of the nonlinearities of the units, the initial state and input govern differential rates of change between the units. When feedback (reset) is incorporated oscillations are produced. By using two of these modules (“sensory” and “motor”) the authors show the network can perform a variety of timing tasks including entrainment and continuation. The authors proceed to compare the performance of the model to previously published human psychophysical data across a range of tasks. The paper provides an elegant, yet fairly simple framework of interval timing that can capture a number of interesting psychophysical findings. Furthermore, the model is fairly unique in that it addresses the problem of sensory

motor timing by proposing the presence of two distinct timing modules that interact and adapt to sensory events. This work provides a significant contribution to the timing field, and has interesting implications for systems level neuroscience.

Most of my comments are minor, with the exception of the Bayesian simulations. It was not clear to me whether it is fair to say the model really captures Bayesian properties of human performance. As I understood it, in Figure 7, the model can be fit to account for the human data by adjusting K , I_0 , and σ . In the model, the Bayesian effect would seem to be almost entirely attributable to changes in I_0 that are left over from the previous trial. So the model is not really taking-in the distribution, but just the previous interval. Which on average, I suppose can look Bayesian because the trials are randomly interleaved. Is this the case or is the model actually sensitive to priors $n-1$, $n-2$, $n-3$, or just $n-1$? If the model is just using the previous trial, this should be explained and explicitly stated. If the model is truly sampling many past trials, the authors should explain this mechanism, and plot the values of I_0 across trials.

The presentation of the standard model and equations can be improved. E.g., as the reader tries to quickly mentally make sense of the equations, they do not work (with weights of 1) unless one knows the key weights are set to 6—which is only stated in a single line in the Methods. That line should also be included when equations 1 and 2 are presented in the results.

Figure 2b should be organized as in Fig. 3b. The vertical organization is easier to grasp.

As I understand it there are other parameter regimes in which the input level x frequency relationship can be inverted (e.g., through weaker Input and smaller y_0). Do the authors see the current relationship (high input – slow) dynamics as a prediction of the model?

Figure 6 contrasts the performance of the model on three conditions with human performance. The human data is from previously published data, but data is shown only from a single subject, so the reader does not really know what the average human performance looks like. It would be nice to provide some information about average human performance in the studies examined.

I believe it is the policy of most journals that it is necessary to explicitly state when the presented data was previously published. In the legend of Fig. 7 it should say “behavioral data modified from Ref 7.

Figure 2 should include a panel E showing the relationship between the mean IPI

and the standard deviation of the IPI (whether or not it is a linear relationship).

Reviewer #3:

Remarks to the Author:

In their manuscript, Egger and colleagues present a dynamical, biologically-inspired model of sensorimotor timing, showing how it can explain data from a range of human experiments of e.g. timing and sync tasks. The model is elegant and strikes a good tradeoff between simplicity and capacity to explain complex phenomena. The addition of experimental human data is laudable. The manuscript was very pleasant to read and, to the best of my knowledge, it describes solid science. It is also great that the authors provide code/data. A few key points are missing though that prevent me from providing a final evaluation of the ms. I'd recommend major revisions and I'd be happy to read a revised version of this ms.

I have two major points and several minor ones. Major points. 1) It is unclear how other models fall short of the phenomena described here and why this one model is much better. A supplement could compare how other frameworks perform, even using the data collected by the authors for this ms, together with e.g. Repp's. 2) How scalable is the model with respect to N, i.e. as the number of elements in an isochronous ISI sequence increase? Looking at Fig.3b it would seem (I may be wrong) that as the # of elements increase, parameter I decreases; what happens to I as N tends to infinity?

A few minor comments follow, but I'd ask the authors to please, please include line numbers in the next submission round: In my opinion this makes the reviewers' work much easier.

pg.1: Why adding all the titles and professorships in the first page, correspondence section? Apart from medical journals, this is quite unusual. By adding this info, there is the risk that unequal relations of power in academia may perpetuate: Would the ms be perceived differently if the last author was a junior postdoc without the 4 lines of mini-CV?

pg.1: First line. Is it a hallmark of humans? This reads quite dismissive of animal capacities...

pg.3: 'excitatory tonic input' The authors should explain what this is.

pg.4: last paragraph before new section: Isn't what the authors show statistically a necessary though not sufficient condition for isochronous rhythm production?

pg.4: what do the authors mean by 'evolve'?

pg.5: Maybe I don't understand fig.2a properly, but shouldn't the feedback arrow point directly to 'I'

pg.5: Maybe I don't understand fig.2d properly, but the quantity depicted is not

Mean IPI (unless the dot values are means of means?)

pg.7, caption: 'isochronous stimuli' isn't isochrony the property of a sequence rather than an individual stimulus?

pg.7, caption: 'Each trace represents...' unclear

pg.7, caption: 'of the level':typo

pg.8, fig 4e: shouldn't this (and other distributions in the ms) be treated as circular, both by using rose plots and circular statistics? For instance, in pg.9, first paragraph: were circular stats used?

References: Some refs are missing publication year or journal.

We thank the reviewers for their careful reading of our manuscript and for their insightful comments! We have prepared a revised manuscript that carefully addresses all their concerns. In the following, the reviewers' comments are quoted in **BLUE**, followed by our responses. Consistent with the manuscript, we will use the following abbreviations:

MPM: motor planning module

SAM: sensory anticipation module

ISI: inter-stimulus-interval

IPI: inter-production-interval

Reviewer 1

1 - Mapping of parameter space. There are several parameters of the model that I would like to understand further. The range of I used for the results is very narrow (0.75–0.78). What is the model's behavior outside of this range? I.e., is the relationship to the IPI monotonic over a larger range of I ?

To address this point, we examined the behavior of the MPM across a wider range of I . When I is smaller than ~ 0.2 or larger than ~ 0.8 , the system cannot drive the output to threshold and thus cannot produce any beat. Between these extremes, the IPI has a non-monotonic relationship with I (decreasing when I is between 0.2 and 0.5, and increasing when I is between 0.5 and 0.8). The results are presented in a supplemental figure (Figure S1). It is also included here as Figure R1 for convenience.

What is the noise level in Figs. 1a and 3b (I can't find these values in the text)?

We thank the reviewer for pointing out this oversight. The sigma for Figure 3b was set to 0.01, and there was no noise in Figure 1a. This information has been added to the figure captions.

Does performance fall apart at higher noise levels?

To address this point, we tested MPM as well as the full model across a wider range of noise levels. In the MPM, the IPI variability increased nonlinearly with noise level and reached its asymptotic value when noise level was higher than $\sim 25\%$ of the input level. The results are presented in a new supplementary figure (Figure S2), also included here (Figure R3) for convenience.

In the full model that couples MPM to the SAM, the noise can induce two unrelated effects. First, noise can cause SAM to force a reset before MPM reaches threshold. This would cause the model to skip a beat. Second, larger noise levels can cause random threshold crossings. This would cause extra beats. The results are presented in a new supplementary figure (Figure S6), also included here (Figure R4) for convenience. These observations can serve as model predictions for future behavioral studies.

How important are the initial values of u and v and the 750ms initialization period? The authors cite Wang et al., Nature Neuroscience, 2018, for the motivation for their basic computational module.

The model is relatively robust to the choice of initial values of u and v , as well as the initialization period. This is now demonstrated in the revised manuscript in a new supplemental figure (Figure S4), also included here (Figure R2) for convenience.

Given that this is a pure theory paper, I believe that plotting the phase space of u and v (as in Wang 2018) would be useful here.

We have also revised the main figures to include the phase plane response of the two-neuron model, as suggested by the reviewer.

2 - The interval reproduction tasks. If I understand correctly, only the prediction module is being used for the interval reproduction tasks, with the reproduction time defined as the threshold crossing for the prediction module. I assume this is because, in the full model, the production module requires multiple stimulus repeats to phase-lock appropriately. However, human subjects are able to produce a motor output with the correct timing after only one (or two) ISIs. I believe the authors should specifically address this issue with respect to their model. Furthermore, I believe the definition of t_p used in Fig. 7 should be stated clearly in either the main text or the figure caption.

The reviewer understood correctly that we only used the SAM to assess the model during the interval reproduction task. To clarify this choice, we wish to highlight the fact that the interval reproduction task consists of two epochs. In the first epoch, the system uses the SAM to measure ISIs but does not produce any output. Therefore, in this epoch, both the SAM and MPM only reset when SAM detects a stimulus (the MPM does not generate an output). In other words, the MPM behaves in an identical fashion to the SAM up until the first motor output. This ensures that the two modules are tightly coupled during the measurement, and that the first output generated by the MPM is appropriately phase-locked to the stimulus.

To demonstrate this process, we have added a new simulation of the full model in which the output of the MPM is clamped to the threshold y_o during the measurement epoch. Results of this simulation show that the MPM is phase-locked to the SAM over various ISIs and different numbers of presented stimuli (N). These results are incorporated in the new supplementary figure (Figure S10), also included here (Figure R6).

Minor points:

- Recommend changing "Because y_p is driven by u_p and v_p , this results..." to "Because y_p is driven by u_p and inhibited by v_p , this results..." for clarity.

Done.

- Recommend adding y-axis values to Fig. 2b.

Done.

- Can you expand on why there is no updating of I at S1 for Fig. 3? It's not obvious to me why that would break the necessary adaptation.

We did not consider adjusting I at S1 because it is not until S2 that the system can measure the interval (S1 marks the beginning of the very first interval). We have clarified this in the revised manuscript.

It is of interest to consider whether I at S1 should be adjusted across trials in accordance with the overall prior over the expected intervals (based on previous trials), and that is what we consider in the later parts of the paper where we address Bayesian integration.

- In Fig. 3b, you have a nice dotted line for y_0 . I recommend using a similar dotted line for I_0 . Done.

- The quantitative definition of asynchrony (i.e., the y-axis of the leftmost plot in 5d) is not referenced in the main text or the figure caption. I recommend alerting the reader at some point that this is defined in the Methods as the authors typically do.

Done.

- In Fig. 6, I initially assumed that the flash times in the top row were aligned with the "Position relative to perturbation" x-axes below, which they are not. For clarity, I recommend these be aligned.

Done.

- In 6a, the purple, black, and pink lines are never defined or referenced in the top plot (I understand they correspond to optimal performance). By the way, I assume defining $\Delta IPI/\Delta ISI$ to be zero prior to the perturbation is a convention since $\Delta ISI=0$ so the ratio is undefined.

Thank you for this detailed assessment. We have defined the lines and expressly defined optimal ΔIPI for $\Delta ISI = 0$ as 0.

- In 6a, the human data show an immediate overshoot on the first trial after the perturbation is detected (black dot above the pink line) while the model takes two trials after error detection to overshoot. Is this just because K is not large enough?

As noted by the reviewer, our model can handle different patterns of overshoot using different values of K and α . The data in Figure 6a of the original manuscript was from one particular subject that had a strong early overshoot (Michon 1967). However, reports in the literature indicate that overshoot to step changes differ across subjects, and is, on average, less pronounced. To address this comment, we made two revisions. First, we revised Figure 6a to show the average overshoot across subjects in a recent experiment (Bavassi et al. 2013). Second, we performed additional model simulations of the model to demonstrate how different values of K and α can cause more or less overshoot. The new results are presented in a supplementary figure (Figure S7), also included here (Figure R5) for convenience.

- In 6b–c, the authors note that they remove the “mean asynchrony”. Is this justified? Why is there a mean asynchrony in the model whilst not in the human data? Can the author propose a possible fix for this?

Humans do have mean asynchronies (Thaut et al. 1998; Semjen et al. 1998), and these are commonly removed from the data during analysis (Repp 2002; Repp 2002; Bavassi et al. 2013). To address this point, we have revised the caption of Figure 6 to read “To facilitate the comparison of the model to human behavior in response to the perturbation, we followed the analysis of human behavior and removed the mean asynchrony adopted before the perturbation from the mean responses of the circuit model.”

- The authors indicate that BIAS and VAR are defined in the Methods. Only the former is. We have now included a definition of VAR in the Methods. Thank you.

- This reviewer notes that he is on a paper relevant to the sentence “These models rely on the rich set of dynamics generated by neurons connected in a local circuit to represent the passage of time”. The paper is Depasquale et al., PLOS One, 2018 (Fig. 7 is the relevant figure). Thank you for pointing this out. We have added the reference.

- This reviewer notes that he is on a paper relevant to the sentence “Given their proposed function as a competitive selection mechanism, these inhibitory pathways may be the substrate for implementing the mutual inhibitory interactions needed for the temporal control of movements”. The paper is Murray and Escola, 2017 (which the authors cite elsewhere; the paper is entirely about mutual inhibition for the temporal control of the motor system). Thank you for pointing this out. We have added the reference.

Reviewer #2

Most of my comments are minor, with the exception of the Bayesian simulations. It was not clear to me whether it is fair to say the model really captures Bayesian properties of human performance. As I understood it, in Figure 7, the model can be fit to account for the human data by adjusting K , I_0 , and σ . In the model, the Bayesian effect would seem to be almost entirely attributable to changes in I_0 that are left over from the previous trial. So the model is not really taking-in the distribution, but just the previous interval. Which on average, I suppose can look Bayesian because the trials are randomly interleaved. Is this the case or is the model actually sensitive to priors $n-1$, $n-2$, $n-3$, or just $n-1$? If the model is just using the previous trial, this should be explained and explicitly stated. If the model is truly sampling many past trials, the authors should explain this mechanism, and plot the values of I_0 across trials.

In general, Bayesian inference involves two computations operating at two different timescales: (1) on a longer timescale (across trials), the observer has to use sample statistics to update its prior belief about statistical regularities in the environment, and (2) on a shorter timescale (every trial), it has to integrate its belief with incoming sensory evidence. The reviewer comments pertains to the former computation: using samples across trials to adaptively update prior expectations. We fully agree. Indeed, in natural settings, environmental statistics are volatile and learning is an integral part of the inference process.

However, the situation is simpler when the statistics of the environment do not change. In this case, the entire Bayesian inference process reduces to a deterministic computation that maps a noisy measurement to an optimal estimate. Several studies have made this point previously, both in Bayesian timing tasks (Jazayeri and Shadlen 2010; Shi and Burr 2016; Egger and Jazayeri 2018), and in other perceptual domains (Oruç et al. 2003; Maloney and Mamassian 2009; Ma and Jazayeri 2014). In our current manuscript, we do not change the prior distribution of samples across trials, and therefore, our focus was on adjusting the model parameters so that it can perform the correct deterministic computation; i.e. map noisy measurements to a Bayes-optimal estimate.

To address the reviewer's comment, we have revised the relevant section of the manuscript and clarified that our modeling of Bayesian inference is based on the assumption that the statistics of the environment are fixed, and that we do not explicitly tackle the problem of learning (i.e., parameter optimization) across-trials that the brain would need to solve when the environment is volatile.

The presentation of the standard model and equations can be improved. E.g., as the reader tries to quickly mentally make sense of the equations, they do not work (with weights of 1) unless one knows the key weights are set to 6—which is only stated in a single line in the Methods. That line should also be included when equations 1 and 2 are presented in the results.
Done.

Figure 2b should organized as in Fig. 3b. The vertical organization is easier to grasp.

Done.

As I understand it there are other parameter regimes in which the input level x frequency relationship can be inverted (e.g., through weaker Input and smaller y_0). Do the authors see the current relationship (high input – slow) dynamics as a prediction of the model?

For our model, it is important that the mutual inhibition operates in a regime with two stable fixed points and one saddle point. In this regime, lower inputs lead to faster speed and vice versa. This can be readily verified by an examination of the phase plane of the two-neuron model. We apologize for not having included this in the original manuscript, and have remedied the problem in the revised version; i.e., the revised Figure 1 includes the full phase plane of the model.

However, as the reviewer has noted, it is possible that the interaction between the input and the dynamics may place the system in other operating regimes. To examine these dependencies, we quantified the relationship between the input level, I , and the IPI across a wide range of inputs. Our work focused on the regime in which IPI increases monotonically with the input, which occurs when I is between 0.5 and 0.788. However, as the reviewer surmised, this relationship is inverted when I is between 0.2 and 0.5. These results are summarized in a new supplemental figure (Figure S1), and in this document (Figure R1) for convenience.

For completeness, we also analyzed the phase plane of the model in low- I ($0.2 < I < 0.5$) and high- I ($I > 1.0$) regimes. These results are shown in a new supplemental figure (Figure S5), and here (Figure R7). Our model can accommodate the low- I regime straightforwardly if we change the sign of the I_s update rule:

$$\tau \frac{dI_s}{dt} = -sK(y_s - y_0)$$

We demonstrate that, after this modification, the SAM performs the update after each stimulus as expected (Figure S5c).

In the high- I regime, the model has only one fixed point on the $u = v$ line (Figure S5). Therefore, unlike the other two regimes in which the system gradually moves away from the $u = v$ line, in this regime, the system will decay toward the $u = v$ line. Again, our model can readily handle this regime if instead of an upper threshold (e.g., $y_0 = 0.7$), we devise a low threshold (e.g., $y_0 = 0.2$) so that the model output moves toward the threshold from above. With this modification, the input-speed relationship is monotonic - higher I leads to shorter IPIs (Figure S5b), and the model can function as expected (Figure S5d).

As the reviewer has noted, these different regimes make distinct predictions about the underlying neurophysiology. The regime which we focused on ($0.5 < I < 0.788$) and the low- I ($0.2 < I < 0.5$) regime both predict that the fast and slow neural trajectories in the state space would be parallel. This is consistent with observations we made previously in our physiology experiments in monkeys (Wang et al. 2018; Remington et al. 2018; Sohn et al. 2019; Egger et

al. 2019). Future experiments can distinguish between these possibilities experimentally by dissociating between excitatory and inhibitory inputs and determining whether larger inputs cause faster or slower neural trajectories.

Figure 6 contrasts the performance of the model on three conditions with human performance. The human data is from previously published data, but the data is shown only from a single subject, so the reader does not really know what the average human performance looks like. It would be nice to provide some information about average human performance in the studies examined.

Thank you for this comment. In the original manuscript, only data from Figure 6a was from a single subject. Figures 6b and c were from data averaged across subjects. In the revised manuscript, we have included average human responses for Figure 6a and we have added language to the caption for Figure 6 to clarify that the data comes from an average across subjects.

I believe it is the policy of most journals that it is necessary to explicitly state when the presented data was previously published. In the legend of Fig. 7 it should say “behavioral data modified from Ref 7.

Thank you. Done.

Figure 2 should include a panel E showing the relationship between the mean IPI and the standard deviation of the IPI (whether or not it is a linear relationship).

We fully agree. This has been added to revised Figure 2 as an inset to panel d.

Reviewer #3

Major points.

1) It is unclear how other models fall short of the phenomena described here and why this one model is much better. A supplement could compare how other frameworks perform, even using the data collected by the authors for this ms, together with e.g. Repp's.

We fully agree that it is essential to compare models. However, a major challenge has been that models of timing capture the underlying computations at various levels of abstraction using algorithmic approaches that cannot be readily mapped onto how neurons interact. Indeed, a major advance in the timing literature would be to understand how such algorithms can be implemented by neural circuit models. Accordingly, a major contribution of our work is that we provide a plausible neural circuit model that can implement such an updating algorithm. This implementation was crucial for comparing the predictions of our model to neural recording in various sensorimotor timing tasks.

However, to address the reviewer's comment, we sought to compare our neural circuit model to existing algorithmic models. One of the most common algorithmic approaches used in time interval production, reproduction and synchronization is to devise a set of equations that adjust IPI by a term that is proportional to the error between the actual and expected ISI. We simulated this algorithmic model in the stimulus perturbation experiments, 1-2-Go and 1-2-3-Go tasks, and Bayesian synchronization/continuation task. The performance of our model is comparable to the algorithmic model in the stimulus perturbation tasks and the 1-2-Go/1-2-3-Go tasks. However, it captures human behavior more accurately in the Bayesian synchronization/continuation task. The superiority of the model in the latter case arises from the a specific architectural feature of our model – the augmented input pathway (Figure 5a, cyan) – which carries an analog signal between different model components. This component is absent in algorithmic models. We have presented these results in three new supplemental figures (Figure S8, S11, and S13), and also here (Figures R8, R9, and R10) for convenience.

2) How scalable is the model with respect to N , i.e. as the number of elements in an isochronous ISI sequence increase? Looking at Fig.3b it would seem (I may be wrong) that as the # of elements increase, parameter I decreases; what happens to I as N tends to infinity?

The model will generalize completely to any N ISIs in a sequence. The value of I will tend towards the value that will generate IPIs that closely match the ISI, up to the influence of noise, not towards zero or infinity. The results in Figure 3b demonstrate two cases, one where the value of I decreases (Figure 3b, bottom left) to match an ISI that was shorter than the IPI produced by I_0 ; (2) where the value of I increases (Figure 3b, bottom right) to match an ISI that was longer than that produced by I_0 . This assertion is backed up by the results of Figure 4c, which demonstrate that the circuit will successfully track 100 ISIs in a sequence. We apologize for our confusing presentation of Figure 3b, which gives the misleading impression that I is decreasing to zero (bottom left) while I_0 is increasing (bottom right). However, it is important to note that I_0 does not change within any trial throughout the paper; the bottom right of Figure 3b

plots I over time in response to a long ISI. To clarify, we have modified Figure 3b to make clear that the bottom panels show that I can increase, decrease (or stay the same) over time while I_0 is constant.

Minor points.

I'd ask the authors to please, please include line numbers in the next submission round: In my opinion this makes the reviewers' work much easier.

Apologies for not including line numbers. This is taken care of.

pg.1: Why adding all the titles and professorships in the first page, correspondence section? Apart from medical journals, this is quite unusual. By adding this info, there is the risk that unequal relations of power in academia may perpetuate: Would the ms be perceived differently if the last author was a junior postdoc without the 4 lines of mini-CV?

This was certainly not our intention. We have fixed this problem.

pg.1: First line. Is it a hallmark of humans? This reads quite dismissive of animal capacities...

Agreed. We have changed the wording '...hallmark of human and animal behavior...' accordingly.

pg.3: 'excitatory tonic input' The authors should explain what this is.

We have elaborated by clarifying that tonic means constant. Excitatory and input are self-explanatory.

pg.4: last paragraph before new section: Isn't what the authors show statistically a necessary though not sufficient condition for isochronous rhythm production?

The increase in variability with larger IPI is a common property of any model in which the effect of noise is accumulated over time. In our model, this effect results from the fact that larger inputs slow the system's dynamics and allow noise to impact the behavior for a longer period of time causing more output variability. We highlighted this feature in the model because this is a hallmark of timing behavior in humans. We have added a sentence to the Results noting that this effect is caused by the reduction in speed (i.e., slower integration toward the threshold).

pg.4: what do the authors mean by 'evolve'?

By 'evolve', we meant change over time toward a target state. This word is used commonly in the literature related to neural dynamics and appears multiple times in our manuscript. To clarify what we mean, we have added an explicit definition (change over time toward a target state) in its first appearance (line 77).

pg.5: Maybe I don't understand fig.2a properly, but shouldn't the feedback arrow point directly to 'I'

The feedback acts as an inhibitory input into u and an excitatory input into v (equations 4 and 5). We chose not to make this explicit in Figure 2a for the purpose of preserving the visual clarity of

the circuit diagram for this figure, Figure 5a, and Figure 6a. We do make this explicit in our model equations.

pg.5: Maybe I don't understand fig.2d properly, but the quantity depicted is not *Mean* IPI (unless the dot values are means of means?)

We apologize for this lack of clarity. We simulated the model once for each input condition. The circles in Figure 2d show the mean of the first 40 IPIs in the simulation, and the errorbars show the corresponding standard deviation over those 40 IPIs. We have clarified this in the Figure caption. To fully address this comment, we have now included a new panel (2e) that shows the mean and standard deviation of IPIs from 100 different simulations at each input level. Evidently, the results are consistent across simulations.

pg.7, caption: 'isochronous stimuli' isn't isochrony the property of a sequence rather than an individual stimulus?

We have changed isochronous to equidistant for clarity.

pg.7, caption: 'Each trace represents...' unclear

We have changed the corresponding text to read 'Each panel contains 100 superimposed lines, each corresponding to the activity of that unit in a different trial.'

pg.7, caption: 'of the level':typo

Corrected. Thank you.

pg.8, fig 4e: shouldn't this (and other distributions in the ms) be treated as circular, both by using rose plots and circular statistics? For instance, in pg.9, first paragraph: were circular stats used?

Thank you for pointing out this oversight. We now report the results of a Rayleigh test of uniformity before and after implementing the augmented input for Figs 4e and 5c. Measures of IPI or asynchrony are not circular by definition.

References: Some refs are missing publication year or journal.

Fixed.

Reviewers' full comments

Reviewer #1

In the paper “A neural circuit model for human sensorimotor timing”, the authors present a model that can produce temporally precise intervals that match the intervals of a stimulus. This work addresses an important outstanding problem in computational neuroscience – how do neural circuits generate temporally precise outputs and adapt to a changing world? As the authors note, this problem has been studied extensively with multiple suggested solutions (e.g., accumulators, oscillators, RNNs) though each has limitations (e.g., unclear neural implementation, inflexibility, etc.). The authors' proposed model, on the other hand, is able to account for a wide range of phenomena and could be implemented neurally.

Specifically, the authors propose a module of two mutually inhibiting units, u and v , which both receive a common excitatory input I and compete with each other via winner-take-all dynamics. The strength of I sets the time-scale of this competition. Units u and v both project to an output y , the former with an excitatory and the latter with an inhibitory connection. The interval measured by the module is given as the time when y crosses some threshold. The speed of this occurring is set by the time-scale of resolving the competition between u and v , which in turn is set by the input I . Thus, a specific I is mapped to a specific interval. After the interval, u , v , and y are reset and the dynamics are repeated.

The authors employ the same basic module twice: once to predict the timing of a stimulus that is assumed to be periodic, and the other to produce outputs. The authors then show that: (1) by using the discrepancy, at the stimulus times, between the y of the prediction module and the threshold, the input I can be updated to improve future predictions; and (2) by using the discrepancy between the y 's of the prediction and production modules, the input I can be updated to obtain phase-locking between the outputs and a periodic stimulus.

This full model can be fit to human behavioral data in a series of interval timing tasks (e.g., synchronization, ready-set-go, continuation after synchronization) and a good match to the behavior can be found. The authors can then study the parameters of the model and make inferences as to the sources of behavioral variability in human subjects.

Overall, this work contributes novel ideas to the field and merits publication. I have only two concerns about which I would like to query the authors.

1 - Mapping of parameter space. There are several parameters of the model that I would like to understand further. The range of I used for the results is very narrow (0.75–0.78). What is the model's behavior outside of this range? I.e., is the relationship to the IPI monotonic over a larger range of I ? What is the noise level in Figs. 1a and 3b (I can't find these values in the text)? Does performance fall apart at higher noise levels? How important are the initial values of u and v and the 750ms initialization period? The authors cite Wang et al., Nature Neuroscience, 2018, for the motivation for their basic computational module. Given that this is a pure theory paper, I believe that plotting the phase space of u and v (as in Wang 2018) would be useful here.

2 - The interval reproduction tasks. If I understand correctly, only the prediction module is being used for the interval reproduction tasks, with the reproduction time defined as the threshold crossing for the prediction module. I assume this is because, in the full model, the production module requires multiple stimulus repeats to phase-lock appropriately. However, human subjects are able to produce a motor output with the correct timing after only one (or two) ISIs. I believe the authors should specifically address this issue with respect to their model. Furthermore, I believe the definition of t_p used in Fig. 7 should be stated clearly in either the main text or the figure caption.

Minor points:

- Recommend changing “Because y_p is driven by u_p and v_p , this results...” to “Because y_p is driven by u_p and inhibited by v_p , this results...” for clarity.
- Recommend adding y-axis values to Fig. 2b.
- Can you expand on why there is no updating of I at $S1$ for Fig. 3? It’s not obvious to me why that would break the necessary adaptation.
- In Fig. 3b, you have a nice dotted line for y_0 . I recommend using a similar dotted line for I_0 .
- The quantitative definition of asynchrony (i.e., the y-axis of the leftmost plot in 5d) is not referenced in the main text or the figure caption. I recommend alerting the reader at some point that this is defined in the Methods as the authors typically do.
- In Fig. 6, I initially assumed that the flash times in the top row were aligned with the “Position relative to perturbation” x-axes below, which they are not. For clarity, I recommend these be aligned.
- In 6a, the purple, black, and pink lines are never defined or referenced in the top plot (I understand they correspond to optimal performance). By the way, I assume defining $\Delta IPI/\Delta ISI$ to be zero prior to the perturbation is a convention since $\Delta ISI=0$ so the ratio is undefined.
- In 6a, the human data show an immediate overshoot on the first trial after the perturbation is detected (black dot above the pink line) while the model takes two trials after error detection to overshoot. Is this just because K is not large enough?
- In 6b–c, the authors note that they remove the “mean asynchrony”. Is this justified? Why is there a mean asynchrony in the model whilst not in the human data? Can the author propose a possible fix for this?
- The authors indicate that BIAS and VAR are defined in the Methods. Only the former is.
- This reviewer notes that he is on a paper relevant to the sentence “These models rely on the rich set of dynamics generated by neurons connected in a local circuit to represent the passage of time”. The paper is Depasquale et al., PLOS One, 2018 (Fig. 7 is the relevant figure).
- This reviewer notes that he is on a paper relevant to the sentence “Given their proposed function as a competitive selection mechanism, these inhibitory pathways may be the substrate for implementing the mutual inhibitory interactions needed for the temporal control of movements”. The paper is Murray and Escola, 2017 (which the authors cite elsewhere; the paper is entirely about mutual inhibition for the temporal control of the motor system).

It was a pleasure to read and review this paper. Thank you for the opportunity.

Sean Escola, M.D., Ph.D.
Assistant Professor of Psychiatry
Center for Theoretical Neuroscience
Columbia University

Reviewer #2

In this manuscript authors present a model of the interaction between sensory and motor timing, and how motor timing can be entrained by rhythmic sensory stimuli. The core of the model is a pair of mutually inhibitory units that are driven by a shared input, and an output unit driven by the difference of the two recurrent units. Because of the nonlinearities of the units, the initial state and input govern differential rates of change between the units. When feedback (reset) is incorporated oscillations are produced. By using two of these modules (“sensory” and “motor”) the authors show the network can perform a variety of timing tasks including entrainment and continuation. The authors proceed to compare the performance of the model to previously published human psychophysical data across a range of tasks. The paper provides an elegant, yet fairly simple framework of interval timing that can capture a number of interesting psychophysical findings. Furthermore, the model is fairly unique in that it addresses the problem of sensory motor timing by proposing the presence of two distinct timing modules that interact and adapt to sensory events. This work provides a significant contribution to the timing field, and has interesting implications for systems level neuroscience.

Most of my comments are minor, with the exception of the Bayesian simulations. It was not clear to me whether it is fair to say the model really captures Bayesian properties of human performance. As I understood it, in Figure 7, the model can be fit to account for the human data by adjusting K , I_0 , and σ . In the model, the Bayesian effect would seem to be almost entirely attributable to changes in I_0 that are left over from the previous trial. So the model is not really taking-in the distribution, but just the previous interval. Which on average, I suppose can look Bayesian because the trials are randomly interleaved. Is this the case or is the model actually sensitive to priors $n-1$, $n-2$, $n-3$, or just $n-1$? If the model is just using the previous trial, this should be explained and explicitly stated. If the model is truly sampling many past trials, the authors should explain this mechanism, and plot the values of I_0 across trials.

The presentation of the standard model and equations can be improved. E.g., as the reader tries to quickly mentally make sense of the equations, they do not work (with weights of 1) unless one knows the key weights are set to 6—which is only stated in a single line in the Methods. That line should also be included when equations 1 and 2 are presented in the results.

Figure 2b should be organized as in Fig. 3b. The vertical organization is easier to grasp.

As I understand it there are other parameter regimes in which the input level \times frequency relationship can be inverted (e.g., through weaker Input and smaller y_0). Do the authors see the current relationship (high input – slow) dynamics as a prediction of the model?

Figure 6 contrasts the performance of the model on three conditions with human performance. The human data is from previously published data, but data is shown only from a single subject, so the reader does not really know what the average human performance looks like. It would be nice to provide some information about average human performance in the studies examined.

I believe it is the policy of most journals that it is necessary to explicitly state when the presented data was previously published. In the legend of Fig. 7 it should say “behavioral data modified from Ref 7.

Figure 2 should include a panel E showing the relationship between the mean IPI and the standard deviation of the IPI (whether or not it is a linear relationship).

Reviewer #3

In their manuscript, Egger and colleagues present a dynamical, biologically-inspired model of sensorimotor timing, showing how it can explain data from a range of human experiments of e.g. timing and sync tasks. The model is elegant and strikes a good tradeoff between simplicity and capacity to explain complex phenomena. The addition of experimental human data is laudable. The manuscript was very pleasant to read and, to the best of my knowledge, it describes solid science. It is also great that the authors provide code/data. A few key points are missing though that prevent me from providing a final evaluation of the ms. I'd recommend major revisions and I'd be happy to read a revised version of this ms.

I have two major points and several minor ones. Major points. 1) It is unclear how other models fall short of the phenomena described here and why this one model is much better. A supplement could compare how other frameworks perform, even using the data collected by the authors for this ms, together with e.g. Repp's. 2) How scalable is the model with respect to N , i.e. as the number of elements in an isochronous ISI sequence increase? Looking at Fig.3b it would seem (I may be wrong) that as the # of elements increase, parameter I decreases; what happens to I as N tends to infinity?

A few minor comments follow, but I'd ask the authors to please, please include line numbers in the next submission round: In my opinion this makes the reviewers' work much easier.

pg.1: Why adding all the titles and professorships in the first page, correspondence section?

Apart from medical journals, this is quite unusual. By adding this info, there is the risk that unequal relations of power in academia may perpetuate: Would the ms be perceived differently if the last author was a junior postdoc without the 4 lines of mini-CV?

pg.1: First line. Is it a hallmark of humans? This reads quite dismissive of animal capacities...

pg.3: 'excitatory tonic input' The authors should explain what this is.

pg.4: last paragraph before new section: Isn't what the authors show statistically a necessary though not sufficient condition for isochronous rhythm production?

pg.4: what do the authors mean by 'evolve'?

pg.5: Maybe I don't understand fig.2a properly, but shouldn't the feedback arrow point directly to 'I'

pg.5: Maybe I don't understand fig.2d properly, but the quantity depicted is not *Mean* IPI (unless the dot values are means of means?)

pg.7, caption: 'isochronous stimuli' isn't isochrony the property of a sequence rather than an individual stimulus?

pg.7, caption: 'Each trace represents...' unclear

pg.7, caption: 'of the level':typo

pg.8, fig 4e: shouldn't this (and other distributions in the ms) be treated as circular, both by using rose plots and circular statistics? For instance, in pg.9, first paragraph: were circular stats used?

References: Some refs are missing publication year or journal.

Figure R1. Mean IPI as a function of input. We simulated the MPM with I ranging from 0.2 to 0.8 and $\sigma_n = 0.01$. For $I < 0.212$ and $I > 0.788$, the dynamics of u and v are constrained such that their difference is always less than y_0 and, therefore, cannot drive y to threshold. For $0.212 < I < 0.5$, the mean IPI is monotonic decreasing. For $0.5 < I < 0.788$, the mean IPI is monotonic increasing.

Figure R2. Performance of the SAM is robust to initial conditions. We simulated the SAM using the same set of parameters as Figure 3b ($K = 5, I_o = 0.77, \sigma_n = 0.01$) during the presentation of three beats of an isochronous rhythm. (a) Predicted interval (the interval between the last beat and when y_s crosses y_o) as a function of the initial state of u_o and v_o for ISI=400 ms. (b) Same as a for ISI=1000 ms. The cross indicates the initial states used in the main text ($u_o = 0.7, v_o = 0.2$). (c) Vertical cross-sections of panels a and b at $v_o = 0.2$, error bars represent standard deviation. (d) Horizontal cross-sections of panels a and b at $u_o = 0.7$, error bars represent standard deviation. (e) Predicted intervals of the SAM as a function of the initialization period (mean \pm standard deviation), and an ISI of 400 ms or 1000 ms. Vertical dashed line indicates the initialization period used in the paper (750 ms).

Figure R3. Standard deviation of IPI of the MPM increases with noise level. We simulated the MPM at different levels of σ_n and I_0 . Results for $I_0 = 0.76$ are shown. For σ_n less than 25% of I_0 (vertical dashed line), noise increased monotonically with σ_n . After that point, the dynamics of the MPM become dominated by the noise and variability in the IPI saturates.

Figure R4. Interaction of noise and α results in skipped productions. a) Examples of skipped productions by the full circuit in response to an 800 ms ISI. Lines show the output of the MPM (y_p , red) and the SAM (y_s , purple) for $\alpha = 0.2$ (top) and $\alpha = 0.4$ (bottom) with $\sigma_n = 0.0079$. Vertical dashed lines indicate the times of the stimuli and asterisks indicate the time at which y_p crosses threshold. In each case, noise prevents one threshold crossing before the SAM is reset. b) The number of IPIs relative to ISIs for different values of σ_n and α . We performed 10 simulations of the full circuit model with different values of σ_n (abscissa) and α (colors) in response to 100 isochronous stimuli (ISI=800 ms), and measured the number of IPIs

relative to when $\sigma_n = 0$. For low levels of noise, increasing α results in the circuit missing more productions, as indicated by the decrease in relative number of IPIs produced. As the level of noise increases, the circuit IPI behavior becomes increasingly variable, leading to an increased probability of early productions in addition to skipped productions. As a result, relatively more IPIs are produced when the level of noise is high relative to the circuit without noise.

Figure R5. Overshoot of circuit model IPIs after a step change in ISI for different levels of α . The degree of IPI overshoot after a step change in ISI from 800 ms (blue line) to 1000 ms (pink line) increases with the level of α . To demonstrate this, we simulated the full circuit as in Figure 6a, but changed the level of α from 0.15 (dark gray circles) to 0.5 (light gray circles) while keeping K fixed at 3. The larger value of α leads to a larger overshoot early in the response to a step change in the ISI. This effect arises from the circuit attempting to match the phase of the stimuli, which is also perturbed by the step change in ISI. When α is large the augmented input to the MPM will also be large in amplitude, resulting in an increased response to the phase difference and a longer mean IPI following the step change.

Figure R6. Production by the MPM during the interval reproduction task. We considered an alternative mechanism for the interval reproduction task in which both the SAM and MPM are involved. Here, the output of the MPM is suppressed until after the final stimulus is shown by clamping the output of the units u and v at their respective values when threshold y_0 is reached and setting $\alpha = 0$. After the last stimulus, the MPM is driven by the input of the SAM, and is responsible for producing the final motor output. a) Response of the units of the MPM (red) and SAM (purple) to three equidistant stimuli with an ISI of 1000 ms. Conventions and model parameters are as Figure 3b. b) Production times t_p (mean \pm standard deviation) for two implementations, by our original model with the SAM only (purple, as in Figure 3b), or the alternative model with the SAM and MPM combined (red), for $N = 2$ stimuli (left) and $N = 3$ stimuli (right). Insets: timing difference between production and the expected time of the next stimulus.

Figure R7. Modified SAM for timing anticipation at different input regimes a) Phase plane of the u and v units of the model, showing trajectories taken by the system at two input regimes, low- I ($I < 0.5$, left), and high- I ($I > 1.0$, right). Open circles indicate the initial state, and filled circles represent the terminal fixed points of the system at different input levels b) Mean of the IPIs as a function of the input level, I , of the modified circuit model that operates in the high- I regime. c) Example responses of the SAM units u_s , v_s , y_s and I_s in the low- I regime to three equidistant stimuli with an ISI of 400 ms (left), or 1000 ms (right). Conventions and model parameters are as Figure 3b. The circuit implementation is identical to our main SAM model,

$$\tau \frac{dI_s}{dt} = -sK(y_s - y_0)$$

with the exception of the I update, which is reversed in sign,

Under these conditions, the model generates (1) differences in trajectory speed and (2) parallel neural trajectories, consistent with neural data (Egger et al., 2019) in a similar task. d) Same as c, but for the high- I regime. The circuit implementation is identical to our main SAM model, with the exception of the threshold y_0 , which was changed to 0.2, and the activity of the y unit approaching this threshold from above. Although the circuit successfully matches IPI to the ISI in this regime, it fails to generate parallel neural trajectories.

Figure R8. Comparison of circuit model response to stimulus perturbations to algorithmic models of human behavior. Following Repp (Repp 2005), we modeled the time of the t_{n+1} th motor output according to the anticipated ISI and the asynchrony between sensory input and motor output. Briefly, we set $t_{n+1} = t_n + \beta_{Asynch} a_n + T_{n+1}$. T_{n+1} represents the anticipated ISI between the n th and $n+1$ th stimulus and is calculated according to $T_{n+1} = T_n + \beta_{ISI}(T_n - ISI_n)$. a_n represents the asynchrony between the n th stimulus, m_n , and the n th production, t_n , and was calculated according to $a_n = t_n - m_n$ (see Methods). (a) Comparison of circuit model (circles) and linear algorithm (lines) in response to a step change in the ISI. Different levels of gray correspond to different weighting of errors in anticipated flash timing (K and β_{ISI} for the circuit and algorithmic models, respectively). (b) Circuit and linear algorithm in response to a phase shift. Conventions as in panel a, but gray scale now corresponds to the sensitivity to phase differences controlled by α for the circuit model and β_{Asynch} for the algorithmic model. (c) Circuit and linear algorithm response to stimulus jitter. Conventions as in panel b.

Figure R9. Comparison of circuit model and the linear algorithm behavior fit to human behavior in the 1-2-Go and 1-2-3-Go tasks. (a) Linear behavioral algorithm fit to example subject behavior (see Methods). Convention is as Figure 7b. (b) BIAS and VAR of linear algorithm and subjects. Convention is as Figure 7c. (c) Difference in BIAS and VAR between the two models and human behavior. For each model and each subject, we calculated the

discrepancies $BIAS_{model} - BIAS_{subject}$, or $VAR_{model}^{1/2} - VAR_{subject}^{1/2}$. These discrepancies are not statistically significantly different between the two models (BIAS in 1-2-Go: $z = 1.066$, $p = 0.8$; BIAS in 1-2-3-Go: $z = -1.42$, $p = 0.08$; $VAR^{1/2}$ in 1-2-Go: $z = -0.237$, $p = 0.4$; $VAR^{1/2}$ in 1-2-3-Go: $z = 0$, $p = 0.5$, Wilcoxon signed-rank test)

Figure R10. Comparison of the circuit model and the linear algorithm behavior fit to human behavior in the synchronization/continuation task. In panels a-c, conventions are as Figures 8a-c. (a) Overall BIAS in the synchronization/continuation task. (b) Inter-production interval (IPI) for different ISIs for an example subject (left) and the linear behavioral algorithm fit to the subject's behavior (right; see Methods). (c) Observed and algorithm BIAS for subjects during each phase of the task. (d) BIAS of human subjects (black), the circuit model (green), and the linear behavioral algorithm (blue) in the first, third, and seventh IPIs. Each dot represents an individual subject or the model fit to that subject. Note that the first and third IPIs occur during synchronization while the seventh IPI occurs during continuation. For all three IPIs, the BIAS is not statistically different (n.s.) between the circuit model and human behavior (First IPI, $z = 1.78$, $p = 0.07$, third IPI, $z = -0.52$, $p = 0.6$, seventh IPI, $z = 1.15$, $p = 0.2$, Wilcoxon signed-rank test). In contrast, the BIAS is significantly different between the algorithm and human behavior for the first and third IPIs (asterisks; First IPI, $z = 2.20$, $p < 0.03$, third IPI, $z = 2.20$, $p < 0.03$, seventh IPI, $z = -1.10$, $p = 0.9$, Wilcoxon signed-rank test). These results demonstrate that the augmented input pathway allows the circuit model to capture the pattern of biases observed in human subjects better than previous models that assumed linear corrections to timing errors.

Reviewers' Comments:

Reviewer #1:

Remarks to the Author:

Thank you for addressing my concerns. I have one final question regarding the new figure panel 2e. Behavioral data in timing tasks typically show Weber scaling (a linear relationship between IPI mean and IPI std). What should we conclude from the fact that this model does not seem to show this scaling feature?

I otherwise recommend this article for publication.

Sean Escola, M.D., Ph.D.

Assistant Professor of Psychiatry Center for Theoretical Neuroscience Columbia University

Reviewer #2:

Remarks to the Author:

The authors have done a very good job in addressing my concerns, and those of the other reviewers. And I think this paper will comprise a significant contribution to the timing field.

Reviewer #3:

Remarks to the Author:

I am happy with how the authors have responded to my queries.

We thank the reviewers for acknowledging our efforts to address their comments on our original submission.

Reviewer #1

I have one final question regarding the new figure panel 2e. Behavioral data in timing tasks typically show Weber scaling (a linear relationship between IPI mean and IPI std). What should we conclude from the fact that this model does not seem to show this scaling feature?

As the reviewer has noted, our model does not exhibit the Weber scaling phenomenon that is observed in human timing. In a noisy dynamical system (such as ours), the exact relationship between variability and mean depends on the form of the noise distribution and the dynamics imposed by the system. In our model, we made the simplifying assumptions that the noise has a Gaussian distribution and that the dynamics can be captured by that of a two-neuron model. We think that capturing the Weber scaling phenomenon would require a more realistic model of the noise in the brain and a more accurate circuit model of the underlying recurrent interactions.

To address this comment in the manuscript, we have changed the relevant section of the text as follows: "In humans, IPIs are variable with a standard deviation that increases linearly with the mean (Ivry and Hazeltine 1995). To evaluate IPI variability in the model, we performed simulations in the presence of Gaussian noise (see Methods). When noise levels were not too high (Supplementary Figure 2), the model exhibited a qualitatively similar behavior to that of humans (Figure 2e): IPI variability increased monotonically with IPI (one-tailed F test; $I = 0.75$ to $I = 0.76$: $p = 0.014$, $F(80,70) = 0.60$; $I = 0.76$ to $I = 0.77$: $p < 0.01$, $F(57,70) = 2.95$; $I = 0.77$ to $I = 0.78$: $p < 0.01$, $F(40,57) = 10.93$). However, this relationship was nonlinear in the model, which is unsurprising given the simplicity of our assumptions regarding the nature of noise in the brain and the underlying dynamics."